# MotionAura: Generating High-Quality and Motion Consistent Videos using Discrete Diffusion

**Onkar Kishor Susladkar**[1], **Jishu Sen Gupta**[2], **Chirag Sehgal**[3], **Sparsh Mittal**[4], **Rekha Singhal**[5]
[1]Northwestern University, [1]Yellow.ai, [2]IIT BHU, [3]Delhi Technological University, [4]IIT Roorkee
[5]TCS Research
onkarsus13@gmail.com,chiragsehgal224@gmail.com
jishusen.gupta.mat22@iitbhu.ac.in
sparsh.mittal@ece.iitr.ac.in,rekha.singhal@tcs.com

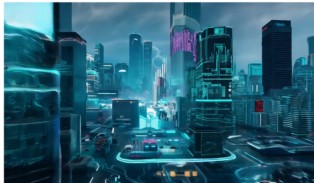
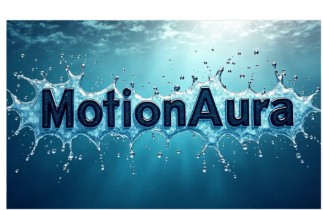
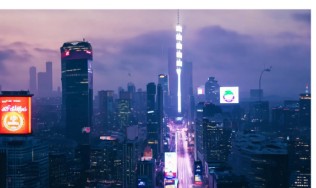

"Futuristic city with neon-lit skyscrapers, flying cars, and a cyberpunk atmosphere, set against a cloudy sky."

"3D 'MotionAura' in ocean tones with dynamic, chaotic motion blur. Hyperkinetic, unsharp edges, high-speed aerial glide over, capturing kinetic energy."

The ambient city noise blends into an electronic soundtrack, enhancing atmosphere. neon glow, dynamic motion, urban symphony, cinematic.

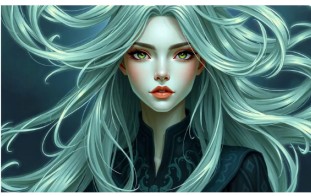
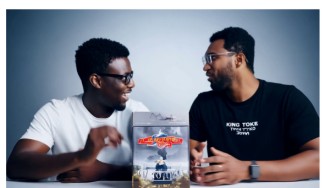
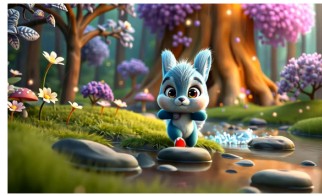

"A female character with long, ethereal hair like the Northern Lights, serene face, and pale skin, dressed in a dark outfit with subtle patterns."

"Two individuals engage in a detailed discussion and unboxing of a limited-edition collectible item, sharing their insights and evaluations in a professional setting."

"A fluffy, blue, rabbit-squirrel hybrid with big eyes explores a vibrant, enchanted forest in high-speed, motion-blurred, hyperkinetic 3D animation, hopping along a sparkling stream with wonder."

Figure 1: We introduce MotionAura, a novel Text-to-Video generation model that predicts discrete tokens obtained from our large scale pre-trained 3D VAE. The displayed frames represent videos generated by our model when provided with the captions shown below each frame. The following link hosts the above generated videos along with other samples.

## ABSTRACT

The spatio-temporal complexity of video data presents significant challenges in tasks such as compression, generation, and inpainting. We present four key contributions to address the challenges of spatiotemporal video processing. First, we introduce the 3D Mobile Inverted Vector-Quantization Variational Autoencoder (3D-MBQ-VAE), which combines Variational Autoencoders (VAEs) with masked token modeling to enhance spatiotemporal video compression. The model achieves superior temporal consistency and state-of-the-art (SOTA) reconstruction quality by employing a novel training strategy with full frame masking. Second, we present MotionAura, a text-to-video generation framework that utilizes vector-quantized diffusion models to discretize the latent space and capture complex motion dynamics, producing temporally coherent videos aligned with text prompts. Third, we propose a spectral transformer-based denoising network that processes video data in the frequency domain using the Fourier Transform. This method effectively captures global context and long-range dependencies for high-

quality video generation and denoising. Lastly, we introduce a downstream task of Sketch Guided Video Inpainting. This task leverages Low-Rank Adaptation (LoRA) for parameter-efficient fine-tuning. Our models achieve SOTA performance on a range of benchmarks. Our work offers robust frameworks for spatiotemporal modeling and user-driven video content manipulation. We release the code, datasets, and models in open-source link .

# 1 INTRODUCTION

Video generation refers to generating coherent and realistic sequences of frames over time, matching the consistency and appearance of real-world videos. Recent years have witnessed significant advances in video generation based on learning frameworks ranging from generative adversarial networks (GANs) (Aldausari et al., 2022), diffusion models (Ho et al., 2022a), to causal autoregressive models (Gao et al., 2024; Tudosiu et al., 2020; Yu et al., 2023c). Video generative models facilitate a range of downstream tasks such as class conditional generation, frame prediction or interpolation (Ming et al., 2024), conditional video inpainting (Zhang et al., 2023; Wu et al., 2024) and video outpainting (Dehan et al., 2022). Video generation has several business use cases, such as marketing, advertising, entertainment, and personalized training.

The video generation process includes two broad tasks. The first task is video frame tokenization, performed using pre-trained variational autoencoders (VAEs) such as 2D VAE used in Stable Diffusion or 3D VQ-VAEs. However, these frameworks fail to achieve high spatial compression and temporal consistency across video frames, especially with increasing dimensions. The second task is visual generation. The early VAE models focused on modeling the continuous latent distribution, which is not scalable with spatial dimension. Recent works have focused on modeling a discrete latent space rather than a continuous one due to its training efficiency. Some works (e.g., Arnab et al. (2021)) train a transformer on discrete tokens extracted by a 3D VQ-VAE framework. However, an autoregressive approach cannot efficiently model a discrete latent space for high-dimensional inputs such as videos.

To address the challenges of visual content generation, we propose *MotionAura*. At the core of MotionAura is our novel VAE, named 3D-MBQ-VAE (3D MoBile Inverted VQ-VAE), which achieves good temporal comprehension for spatiotemporal compression of videos. For the video generation phase, we use a denoiser network comprising a series of proposed Spectral Transformer blocks trained using the masked token modeling approach. Our novelty lies both in the architectural changes in our transformer blocks and the training pipeline. These include using a learnable 2D Fast Fourier Transform (FFT) layer and Rotary Positional Embeddings (Su et al., 2021). Following an efficient and robust, non-autoregressive approach, the transformer block predicts all the masked tokens at once. To ensure temporal consistency across the video latents, during tokenization, we randomly mask one of the frames completely and predict the index of the masked frame. This helps the model learn the frame sequence and improve the temporal consistency of frames at inference time. Finally, we address a downstream task of sketch-guided video inpainting. To our knowledge, ours is the first work to address the sketch-guided video inpainting task. Using our pre-trained noise predictor as a base and the LoRA adaptors (Hu et al., 2021), our model can be finetuned for various downstream tasks such as class conditional video inpainting and video outpainting. Figure 1 shows examples of text-conditioned video generation using MotionAura. Our contributions are:

- We propose a novel 3D-MBQ-VAE for spatio-temporal compression of video frames. The 3D-MBQ-VAE adopts a new training strategy based on the complete masking of video frames. This strategy improves the temporal consistency of the reconstructed video frames.

- We introduce MotionAura, a novel framework for text-conditioned video generation that leverages vector quantized diffusion models. That allows for more accurate modeling of motion and transitions in generated videos. The resultant videos exhibit realistic temporal coherence aligned with the input text prompts.

- We propose a denoising network named spectral transformer. It employs Fourier transform to process video latents in the frequency domain and, hence, better captures global context and long-range dependencies. To pre-train this denoising network, we add contextually rich captions to the WebVID 10M dataset and call this curated dataset WebVid-10M-recaptioned.

- We are the first to address the downstream task of sketch-guided video inpainting. To realize this, we perform parameter-efficient finetuning of our denoising network using LORA adaptors. The experimental results show that 3D-MBQ-VAE outperforms existing networks in terms of reconstruction quality. Further, MotionAura attains state-of-the-art (SOTA) performance on text-conditioned video generation and sketch-guided video inpainting.

- We curated two datasets, with each data point consisting of a Video-Mask-Sketch-Text conditioning, for our downstream task of sketch-guided video inpainting. We utilized YouTube-VOS and DAVIS datasets and captioned all the videos using Video LLaVA-7B-hf. Then, we performed a CLIP-based matching of videos with corresponding sketches from QuickDraw and Sketchy.

## 2 RELATED WORKS ON VISUAL CONTENT GENERATION

**Image generation:** Text-guided image synthesis has resulted in high quality and spatially coherent data (Esser et al., 2024; Sun et al., 2024b). Generative Adversarial Networks (Sauer et al., 2022; Karras et al., 2021; Goodfellow et al., 2014) model visual distributions to produce perceptually rich images but fail to capture the complete spectrum of data distribution. Moreover, these networks are difficult to optimize. Kingma & Welling (2019) introduce another approach to generating visual data wherein the decoders of the autoencoder-like architectures are trained to reconstruct images by sampling over the distributions provided by encoders. This regularization strategy helps to learn continuous and meaningful latent space. Quantization of the produced latents (van den Oord et al., 2018; Susladkar et al., 2025) further improves the quality of generated data using various sampling methods over the learned latent. While existing sampling methods can capture dense representations in their latent space, they fail to produce sharp, high-quality images. Diffusion models train networks to learn a denoising process, appropriately fitting the inductive bias in image data. Recent developments such as DDPMs (Ho et al., 2020), latent diffusion methods (Rombach et al., 2022a) and diffusion transformers (Esser et al., 2024; Peebles & Xie, 2023; Chen et al., 2024; 2023) have revolutionized the field of image generation.

**Video generation:** Video generation is even more challenging than image generation since generating high-quality videos demands both spatial and temporal consistencies. For video generation, researchers have used both transformer models and diffusion models. VideoGPT (Yan et al., 2021) generates videos autoregressively using the latents obtained from the VQ-VAE tokenizer. Processing videos in the latent space rather than the pixel space reduces training and inference latency. The early diffusion-based video generation models transferred weights from the TextToImage (T2I) approaches and adapted them by inflating the convolutional backbone to incorporate the temporal dynamics (Ho et al., 2022b). Latent diffusion models (LDM) based Text2Video (T2V) approaches generate higher-quality videos of longer duration (Blattmann et al., 2023). However, diffusion-based approaches using convolution backbones fail to produce high frame-rate videos and generally rely on other architectures to interpolate intermediate frames. Recent works have combined transformer-based predictions with diffusion-based pretraining (Yu et al., 2023b; 2024; Kondratyuk et al., 2024).

## 3 MOTIONAURA: ARCHITECTURE AND TRAINING

We introduce MotionAura, a novel video generation model designed to produce high-quality videos with strong temporal consistency. At the core of MotionAura is 3D-MBQ-VAE, our novel 3D VAE that achieves high reconstruction quality. The encoder of this pretrained 3D-MBQ-VAE encodes videos into a latent space, forming the first essential component of our approach. The second novelty is a spectral transformer-based diffusion model, which diffuses the encoded latents of videos to produce high-quality videos. Section 3.1 discusses the large-scale pretraining of 3D-MBQ-VAE for learning efficient discrete representations of videos. Section 3.2 discusses pretraining of the noising predictor, whereas Section 3.3 presents the Spectral Transformer module for learning the reverse discrete diffusion process.

### 3.1 PRE-TRAINING OF 3D-MBQ-VAE

We employed a 3D-VQ-VAE (Vector Quantized Variational Autoencoder) to tokenize videos in 3D space; refer Appendix B.1 for the architecture of 3D-MBQ-VAE. We pretrained our network on

the YouTube-100M dataset to ensure that it produces video-appropriate tokens. The extensive content variety in the YouTube-100M dataset enhances the model's generalization and representational capabilities. Figure 2 shows our pretraining approach for learning efficient latents with temporal and spatial consistencies. The pretraining leverages two approaches for optimal video compression and discretization: (i) Random masking of frames (He et al., 2021) using 16×16 and 32×32 patches. This training ensures self-supervised training of the autoencoder to efficiently learn spatial information. (ii) Complete masking of a single frame in the sequence to enforce learning of temporal information. Let $B$ be the batch size, $C$ be number of channels, $N$ the number of frames per video, $H$, and $W$ be the height and width of the video frames, respectively. For a given video $V \in \mathbb{R}^{(B,N,3,H,W)}$, a randomly selected frame $f_i \in \mathbb{R}^{(3,H,W)}$ is masked completely. The encoder $\mathcal{E}(V)$ returns $z_c \in \mathbb{R}^{(B,C,N,H/16,W/16)}$. This compressed latent in the continuous domain is fed into the MLP layer to return $P(f_i)$, which shows the probability of $i^{th}$ frame being completely masked.

The same $z_c$ is discretized using the Euclidean distance-based nearest neighbor lookup from the codebook vectors. The quantized latent is $z \in \mathbb{C}$ where $\mathbb{C} = \{e_i\}_{i=1}^{K}$. $\mathbb{C}$ is the codebook of size $K$ and $e$ shows the codebook vectors. This quantized latent is reconstructed using our pre-trained 3D-MBQ-VAE decoder $\mathcal{D}(z)$. Eq. 1 shows the combined loss term for training, where sg represents the stop gradient. We include *Masked Frame Index Loss* or $\mathcal{L}_{\text{MFI}}$ in the loss function to learn from feature distributions across the frames. This loss helps VAE to learn frame consistency.

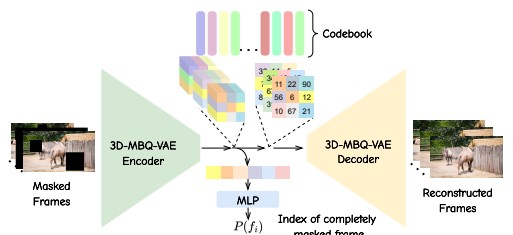

Figure 2: Our proposed pre-training method for 3D-MBQ-VAE architecture

$$\mathcal{L}_{MBQ-VAE} = \underbrace{\|V - \hat{V}\|_2^2}_{\mathcal{L}_{rec}} + \underbrace{\|\text{sg}[\mathcal{E}(V)] - V\|_2^2}_{\mathcal{L}_{codebook}} + \beta \underbrace{\|\text{sg}[V] - \mathcal{E}(V)\|_2^2}_{\mathcal{L}_{commit}} - \underbrace{\log(P(f_i))}_{\mathcal{L}_{MFI}} \quad (1)$$

## 3.2 PRE-TRAINING OF NOISING PREDICTOR FOR TEXT2VIDEO GENERATION

With our pre-trained video 3D-MBQ-VAE encoder $\mathcal{E}$ and codebook $\mathbb{C}$, we tokenize video frames in terms of the codebook indices of their encodings. Figure 3 shows the pretraining approach of the Spectral Transformer ($\epsilon_\theta$). Given a video $V \in \mathbb{R}^{(B,N,3,H,W)}$, the quantized encoding of $V$ is given by $z_0 = \mathbb{C}(\mathcal{E}(V)) \in \mathbb{R}^{(B,N,H/16,W/16)}$. In continuous space, forward diffusion is generally achieved by adding random Gaussian noise to the image latent as a function of the current time step 't'. However, in this discrete space, the forward pass is done by gradually corrupting $z_0$ by masking some of the quantized tokens by a <MASK> token following a probability distribution $P(z_t \mid z_{t-1})$. The forward process yields a sequence of increasingly noisy latent variables $z_1, ..., z_{\mathbb{T}}$ of the same dimension as $z_0$. At timestep $\mathbb{T}$, the token $z_{\mathbb{T}}$ is a pure noise token.

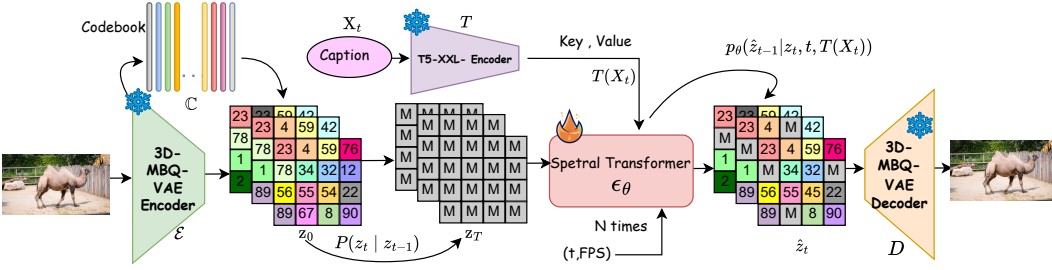

Figure 3: Discrete diffusion pretraining of the spectral transformer involves processing tokenized video frame representations from the 3D-MBQ-VAE encoder. These representations are subjected to random masking based on a predefined probability distribution. The resulting corrupted tokens are then denoised through a series of $N$ Spectral Transformers. Contextual information from text representations generated by the T5-XXL-Encoder aids in this process. The denoised tokens are reconstructed using the 3D decoder

Starting from the noise $z_{\mathbb{T}}$, the reverse process gradually denoises the latent variables and restores the real data $z_0$ by sampling from the reverse distribution $q(z_{t-1}|z_t, z_0)$ sequentially. However, since $z_0$ is unknown in the inference stage, we must train a denoising network to approximate the conditional distribution $\epsilon_\theta(z_t|z_0, t, T(X_t))$. Here, $t$ represents the time step conditioning, which gives information regarding the degree of masking at step $t$. $T(X_t)$ represents the text conditioning obtained by passing the video captions $X_t$ through the T5-XXL (Tay et al., 2022) text encoder $T$. We propose a novel Spectral Transformer block $\epsilon_\theta$ to model the reverse diffusion process (Section 3.3). The objective we maximize during training is the negative log-likelihood of the quantized latents given the text and time conditioning. The likelihood is given by $L_{\text{diff}} = -\log\left(\epsilon_\theta\left(z_t|z_0, t, T(X_t)\right)\right)$.

## 3.3 PROPOSED SPECTRAL TRANSFORMER

We propose a novel Spectral Transformer, which processes video latents in the frequency domain to learn video representations more effectively. Consider a quantized latent $z_t \in \mathbb{R}^{(B,N,H/16,W/16)}$ at time step $t$. First, the latent is flattened to $\mathbb{R}^{(B,N\times H/16\times W/16)}$ which along with time step $t$ and $FPS$ (frames per second), are passed to our proposed Spectral Transformer block. Initially, the passed tokens are fed into an Adaptive Layer Normalization (AdLN) block to obtain $B_t$, where $B_t = \text{AdLN}(z_t, t, \text{FPS})$.

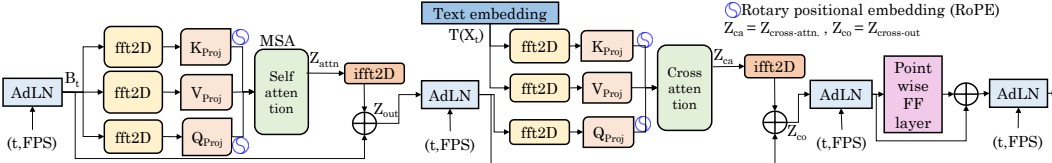

Figure 4: Architecture of spectral transformer

As shown in Figure 4, before computing the self-attention, the latent representation $B_t$ is transformed into the frequency domain using a 2D Fourier Transform Layer (*fft2D*) for each of $\mathbf{K}$, $\mathbf{V}$, and $\mathbf{Q}$. This layer applies two 1D Fourier transforms (Susladkar et al., 2024; 2023) - along the embedding dimension and the sequence dimension thereby converting spatial domain data into the frequency domain. This enables the segregation of high-frequency features from low-frequency ones, enabling more efficient manipulation and learning of spatial features. We take only the real part of *fft2D* layer outputs to make all quantities differentiable (Eq. 2). Here, MSA stands for multi-headed self-attention.

$$\mathbf{Z}_{\text{attn}} = \text{MSA}(\mathbf{Q}_{\text{Proj}}(\mathbb{R}(\textit{fft2D}(Z_t))), \mathbf{K}_{\text{Proj}}(\mathbb{R}(\textit{fft2D}(Z_t))), \mathbf{V}_{\text{Proj}}(\mathbb{R}(\textit{fft2D}(Z_t)))) \tag{2}$$

The frequency-domain representations are then fed into the self-attention block. Using Rotary Positional Embeddings (RoPE) at this stage helps increase the context length of the input and results in faster convergence. The output $\mathbf{Z}_{attn}$ undergoes an Inverse 2D Fourier Transform (*ifft2D*) to revert to the spatial domain. Then, a residual connection with the AdLN layer is applied to preserve the original spatial information and integrate it with the learned attention features. This gives the output $\mathbf{z}_{\text{out}}$ (Eq. 3).

$$\mathbf{z}_{\text{out}} = \text{AdLN}(\mathbb{R}(\textit{ifft2D}(\mathbf{Z}_{\text{attn}})) + \text{AdLN}(\mathbf{z}_t, t, FPS)) \tag{3}$$

Similarly, in the cross-attention computation, the text-embedding $T(X_t)$ is first transformed using learnable Fourier transforms for both $\mathbf{K}$ and $\mathbf{V}$. The $\mathbf{Q}$ representation from the preceding layer is also passed through the learnable Fourier transforms (Eq. 4). These frequency-domain representations are then processed by the text-conditioned cross-attention block.

$$\mathbf{Z}_{\text{cross-attn}} = \text{MSA}\left(\mathbf{Q}_{\text{Proj}}(\mathbb{R}(\textit{fft2D}(z_{\text{out}}))), \mathbf{K}_{\text{Proj}}(\mathbb{R}(\textit{fft2D}(\text{T}(X_t)))), \mathbf{V}_{\text{Proj}}(\mathbb{R}(\textit{fft2D}(\text{T}(X_t))))\right) \tag{4}$$

Then, the frequency domain output is converted back to the spatial domain using *ifft2D* layer, and a residual connection is applied (Eq. 5).

$$\mathbf{z}_{\text{cross-out}} = \text{AdLN}(\mathbb{R}(\textit{ifft2D}(\mathbf{Z}_{\text{cross-attn}})) + \text{AdLN}(\mathbf{z}_{out}, t, FPS)) \tag{5}$$

Finally, an MLP followed by a softmax layer gives the predicted denoised latent given $z_t$. This process is repeated until we get the completely denoised latent $P(z_0 \mid z_t, t, T(X_t))$. Finally, these output latents are mapped back into video domain using our pre-trained 3D-MBQ-VAE decoder $\mathcal{D}$.

## 4 MOTIONAURA FOR SKETCH-GUIDED VIDEO INPAINTING

*MotionAura* can be flexibly adapted to downstream video generation tasks such as sketch-guided video inpainting. In contrast with the previous works on text-guided video inpainting (Zhang et al., 2024; Ceylan et al., 2023), we use both text and sketch to guide the video inpainting. The sketch-text pair makes the task more customizable. For finetuning the model, we use the datasets curated from YouTube-VOS and QuickDraw (Appendix C.4). We now explain the parameter-efficient fine-tuning approach. Figure 5 details the entire process. For a given video $V = \{f_i\}_{i=1}^J$, we obtain a masked video $V_m$ using a corresponding mask sequence $m = \{m_i\}_{i=1}^J$ where $V_m$ is given by $V_m = \{f_i \odot (1 - m_i)\}_{i=1}^J$. We then discretize both these visual distributions using our pretrained 3D-MBQ-VAE encoder, such that $z_m = \mathcal{E}(V_m)$ and $z_0 = \mathcal{E}(V)$.

Given a masked video, our training objective is to obtain the unmasked frames with the context driven by sketch and textual prompts. Hence, we randomly corrupt the discrete latent $z_0$ with <MASK> token (Section 3.2) to obtain the fully-corrupted latent $z_{\mathbb{T}}$. After performing the forward diffusion process, $z_{\mathbb{T}}$ and $z_m$ are flattened and concatenated. This concatenated sequence of discrete tokens is passed through an embedding layer to obtain $Z'_t \in \mathbb{R}^{(B,S,E)}$ where $Z'_t$ represents the intermediate input token sequence.

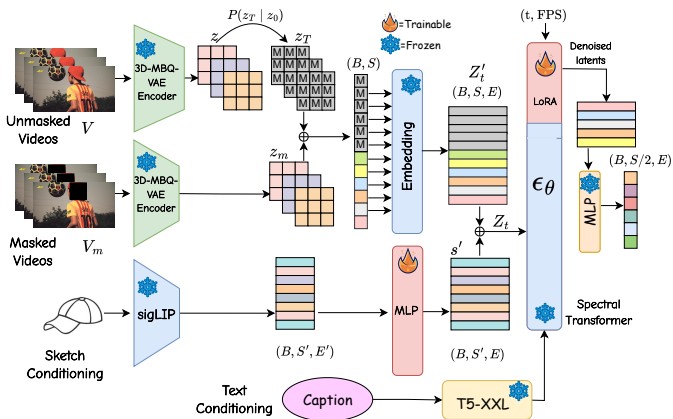

Figure 5: Sketch-guided video inpainting process. The network inputs masked video latents, fully diffused unmasked latent, sketch conditioning, and text conditioning. It predicts the denoised latents using LoRA infused in our pre-trained denoiser $\epsilon_\theta$.

As for the preprocessing of the sketch, we pass the sketch through the pretrained SigLIP image encoder (Zhai et al., 2023) and consecutively through an MLP layer to finally obtain $s' \in \mathbb{R}^{(B,S',E)}$. At last, $s'$ is concatenated with $Z'_t$ to obtain the final sequence of tokens $Z_t$, where $Z_t = Z'_t \oplus S'$ and $Z_t \in \mathbb{R}^{(B,S+S',E)}$. This input sequence is responsible for the above mentioned *context* to the diffusion network. The input sequence already contains the latents of the unmasked regions to provide the available spatial information. It also contains the *sketch* information that helps the model understand and provide desirable visual details.

The base model is frozen during the inpainting task, and only the adapter modules are trained. The pretrained spectral transformer is augmented with two types of LoRA modules during finetuning: (1) Attention LoRA, applied to $K_{proj}$ and $Q_{proj}$ layers of the self-attention and cross-attention layers of the transformer module, operating in parallel to the attention layers, (2) Feed Forward LoRA, applied to the output feed-forward layer in a sequential manner. These two adapter modules share the same architecture wherein the first Fully Connected (FC) layer projects the high-dimensional features into a lower dimension following an activation function. The next FC layer converts the lower dimensional information to the original dimension. The operation is shown as $L'(x) = L(x) + W_{up}(\text{GeLU}(W_{down}(x)))$ such that $W_{up} \in \mathbb{R}^{(d,l)}$ and $W_{down} \in \mathbb{R}^{(l,d)}$ with $l < d$. Here $W_{up}$ and $W_{down}$ represent the LoRA weights, $L'(x)$ represents the modified layer output and $L(x)$ represents the original layer output. The $W_{down}$ is initialized with the distribution of weights of $L(x)$ to maintain coherence with the frozen layer.

## 5 EXPERIMENTAL RESULTS

We now present experimental results. The details of experimental setup are provided in Appendices C and D. Additional qualitative results are provided in Appendices A, G and F.

## 5.1 LATENT RECONSTRUCTION RESULTS OF 3D-MBQ-VAE

For evaluating our 3D-MBQ-VAE, we selected COCO-2017 and WebVID validation datasets. Following Zhao et al. (2024), we crop each frame to 256×256 resolution and sample 48 frames per video sequentially. As shown in Table 1, our proposed 3D-MBQ-VAE consistently outperforms SOTA 3D VAEs across all metrics. This can be attributed to a novel combination of training strategies and the enhanced loss function. Randomly masking parts of various frames He et al. (2021) enables our model to efficiently learn spatial information. Further, inclusion of $\mathcal{L}_{\mathrm{MFI}}$ in the loss function emphasizes learning efficient temporal features. On removing $L_{\mathrm{mfi}}$, LPIPS metric decreases substantially. This highlights its role in preserving perceptual quality. Our 3D-MBQ-VAE model is also capable of supporting joint image and video training, similar to W.A.L.T Gupta et al. (2025) and CV-VAE. As shown in Table 1, joint training provides superior results than video training alone.

Table 1: Quantitative comparison of VAEs for video reconstruction task on COCO-Val and WebVid-Val datasets. Frame compression rate (FCR) is the ratio between the size of video frame before and after compression. Compatibility (comp.) represents if the model can be used as a VAE for existing generative models. Our method demonstrates superior performance on all metrics. All models are trained on videos, except the "3D-MBQ-VAE (Video+Images)" variant that is jointly trained on video and images. Hence, its results are not compared with other methods.

| Method | Params | FCR | Comp. | COCO-Val PSNR(↑) / SSIM(↑) / LPIPS(↓) | WebVid-Val PSNR / SSIM / LPIPS |
|---|---|---|---|---|---|
| VAE-SD2.1 (Rombach et al., 2022b) | 34M + 49M | 1× | - | 26.6 / 0.773 / 0.127 | 28.9 / 0.810 / 0.145 |
| VQGAN (Esser et al., 2021) | 26M + 38M | 1× | × | 22.7 / 0.678 / 0.186 | 24.6 / 0.718 / 0.179 |
| TATS (Ge et al., 2022) | 7M + 16M | 4× | × | 23.4 / 0.741 / 0.287 | 24.1 / 0.729 / 0.310 |
| VAE-OSP (Lab & etc., 2024) | 94M + 135M | 4× | × | 27.0 / 0.791 / 0.142 | 26.7 / 0.781 / 0.166 |
| CV-VAE(2D+3D) (Zhao et al., 2024) | 68M + 114M | 4× | ✓ | 27.6 / 0.805 / 0.136 | 28.5 / 0.817 / 0.143 |
| CV-VAE(3D) (Zhao et al., 2024) | 100M + 156M | 4× | ✓ | 27.7 / 0.805 / 0.135 | 28.6 / 0.819 / 0.145 |
| 3D-MBQ-VAE (Without $L_{\mathrm{mfi}}$) | 140M + 177M | 4× | ✓ | 30.8 / 0.840 / 0.112 | 31.4 / 0.849 / 0.134 |
| 3D-MBQ-VAE | 140M + 177M | 4× | ✓ | **31.2 / 0.848 / 0.092** | **32.1 / 0.858 / 0.108** |
| 3D-MBQ-VAE (Video+Images) | 140M + 177M | 4× | ✓ | 33.0 / 0.866 / 0.087 | 34.2 / 0.877 / 0.092 |

Figure 6 shows the t-SNE plots, where each color represents a different class in the dataset. Evidently, the better clustering and separation by 3D-MBQ-VAE shows that it is most effective in feature extraction and representation learning. The qualitative results in Figure 7 further confirm that 3D-MBQ-VAE provides reconstructions of highest quality and consistency.

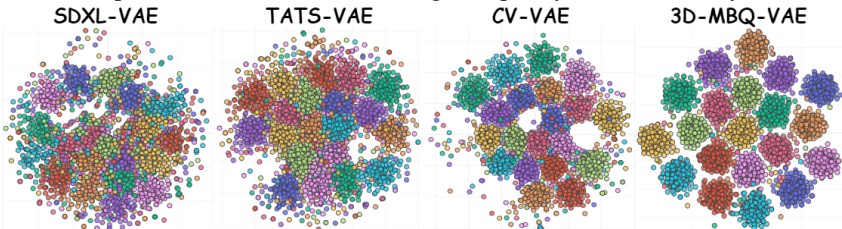

Figure 6: t-SNE plots showing the representations learned by various VAEs.

## 5.2 RESULTS OF TEXT CONDITIONED VIDEO GENERATION

For evaluation purposes, we create three variants of our model, Small (0.89B parameters), Medium (1.94B parameters), and Large (3.12B parameters) as shown in Table 2. Here, blocks refers to the number of spectral transformer blocks ($N$ in Figure 3).

Table 2: Comparison of MotionAura variants.

| Model | Embedding Size | Attn head | Blocks |
|---|---|---|---|
| S | 1024 | 16 | 17 |
| M | 2048 | 16 | 28 |
| L | 4096 | 32 | 37 |

For zero-shot Text2Video generation, we pre-train our Text2Video models with specific configurations (refer appendix D.3). We compare our model with AnimateDiff (Guo et al., 2024), CogVideoX-5B (Yang et al., 2024a) and SimDA (Xing et al., 2023). The texts given as text conditioning to all these models are taken from the WebVID 10M dataset (with a maximum length of 180). However, we observe that the captions in the original WebVID 10M dataset lacked contextual richness, which limits the scope of generation. Hence, we propose our WebVID 10M-*recaptioned* dataset with contextually rich textual data (details in Appendix C). We evaluate the pre-trained models on the MSR-VTT dataset (Chen et al., 2022) using standard metrics such as FVD and CLIPSIM. We also included two human assessment-based metrics, viz., Motion Quality (MQ) and Temporal Consistency (TC) (Liu et al., 2024; Feng et al., 2024).



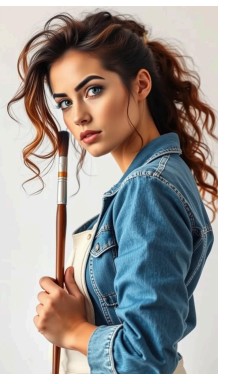
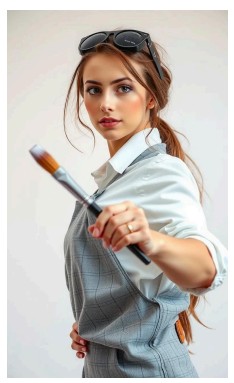

**MotionAura-L**  **MotionAura-L(w/o fft)**

Figure 7: Comparison of 3D-MBQ-VAE (ours) with existing VAEs after reconstruction from Stable diffusion-XL produced latent. Zooming in on the figure reveals significant differences in image quality, fidelity to the original latent representations, and the preservation of fine details.

*A female painter with a brush in hand, white background, painting, looking very powerful.*

Figure 8: The videos corresponding to these frames show the improvement in motion quality brought by FFT layers. For videos, click on link.

The EvalCrafter benchmark helps us quantitively evaluate the quality of video generation on these aspects. The framework also aligns objective metrics with human opinions to enhance evaluation accuracy. 100 participants ranked each of the $\mathbb{Y}$ models such that the model ranking first received $\mathbb{Y} - 1$ points and the model ranking last received 0 points.

As shown in Table 3, pretraining on our newly curated dataset improves the performance of all the models, highlighting the importance of enriched captions. Further, MotionAura-L without the use of *fft2D* and *ifft2D* layers has inferior performance than the full MotionAura-L network, showcasing the effectiveness of our proposed Spectral Transformer architecture. Finally, our largest model, **MotionAura-L**, achieves the best results across all metrics, demonstrating its superior capacity for capturing both motion dynamics and temporal consistency.

We measure the average inference latency over text sample lengths of 30 to 180 tokens. For generating a 5-second video, MotionAura-L takes 38 seconds, compared to 41 seconds of CogVideoX-5B (Table 3). To reduce the latency, the FFT layers can be skipped, or the small/medium variants can be used. Notably, MotionAura-S is comparable in latency to AnimateDiff and superior in performance metrics. MotionAura can generate videos of up to 10 seconds, whereas previous works generate up to 6 seconds videos. MotionAura-L takes 83 seconds to generate a 10-second video.

Table 3: Results of the text-conditioned video generation (Text2Video) models. For all the techniques, evaluation was done on MSR-VTT dataset. Recaptioning the dataset improves all the metrics. Inference Time was calculated on a single A100.

| Method | Params | Train: WebVID 10M (FVD↓/CLIPSIM↑/MQ↑/TC↑) | WebVid 10M - recaptioned (FVD/CLIPSIM/MQ/TC) | Inf Tims(s) |
|---|---|---|---|---|
| SimDA | 1.08B | 456 / 0.1761 / 65.28 / 195 | 433 / 0.1992 / 67.82 / 196 | 8.6 |
| AnimateDiff | 1.96B | 402 / 0.2011 / 68.97 / 197 | 379 / 0.2201 / 69.78 / 199 | 11.1 |
| CogVideoX-5B | 5.00B | 380 / 0.2211 / 73.37 / 203 | 357 / 0.2429 / 75.51 / 205 | 41.0 |
| MotionAura-L (w/o fft) | 3.12B | 379 / 0.2201 / 73.44 / 202 | 351 / 0.2441 / 75.87 / 203 | 33.4 |
| MotionAura-S | 0.89B | 391 / 0.2104 / 70.29 / 199 | 364 / 0.2303 / 71.28 / 200 | 12.0 |
| MotionAura-M | 1.94B | 383 / 0.2207 / 72.37 / 200 | 360 / 0.2333 / 73.47 / 202 | 20.0 |
| MotionAura-L | 3.12B | **374 / 0.2522 / 74.59 / 204** | **344 / 0.2822 / 76.62 / 207** | 38.0 |

Figure 8 shows sample frames from videos generated by the full MotionAura-L and MotionAura-L(without fft2d). The videos highlight superior motion quality achieved on using the FFT layers. Figure 9 compares our model with CogVideoX-5B and AnimateDiff. Both quantitative and qualitative results highlight the superior performance of our model, which can be attributed to several key innovations. AnimateDiff relies on the *image VAE* from Stable Diffusion to encode video latents, which limits their representational power compared to the latents produced by our 3D-MBQ-VAE. AnimateDiff inflates a 2D diffusion model into 3D using a temporal sub-module for video, but this approach fails to ensure true temporal coherence, as seen when comparing its results to those of MotionAura. CogVideoX-5B employs a *video VAE* in continuous space. This approach struggles

with efficiently representing the high dimensionality of video frames, making it less effective than discrete space representations. Additionally, our use of spectral transformer blocks, with integrated 2D Fourier transforms, enhances both vision and language modeling by capturing complex spatial and temporal patterns. Replacing traditional sinusoidal embeddings with RoPE embeddings further improves the handling of longer video sequences. These architectural improvements collectively improve our overall performance.

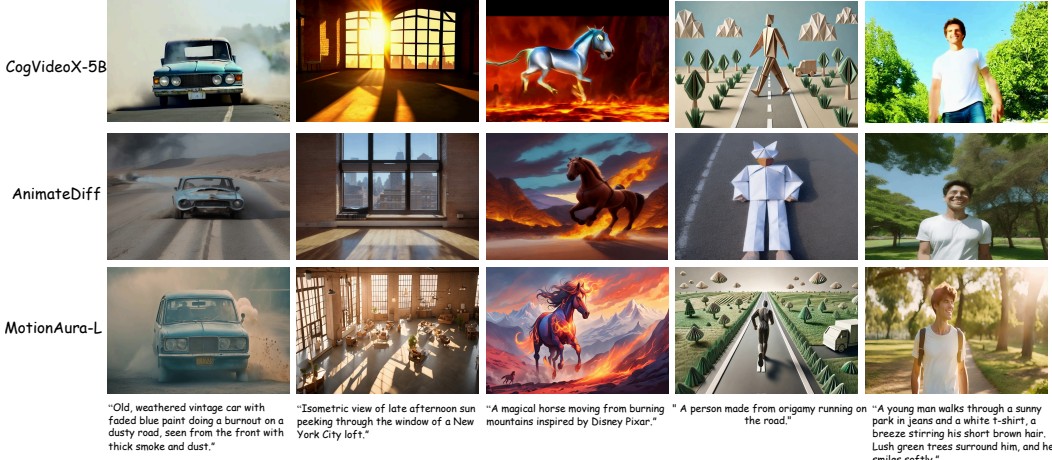

Figure 9: Text-conditioned video generation results. Our model shows superior temporal consistency and generation quality. Click on the link to view the videos.

## 5.3 RESULTS OF SKETCH GUIDED VIDEO INPAINTING

We now assess the models' performance on the newly introduced task of sketch-guided video inpainting. As outlined in Section 4, our approach adapts zero-shot Text2Video models using LoRA modules. For a fair comparison with CogVideoX-5B, we adapted it in a similar manner as our proposed approach. Both existing and our models were pre-trained on the WebVid 10M-*recaptioned* dataset and evaluated on two newly curated datasets, based on existing video (YouTube-VOS and DAVIS) and sketch (QuickDraw) datasets. These datasets have been specifically designed to accommodate the sketch-based inpainting task, featuring four key components for each video sample: textual prompts, sketches, segmentation masks, and raw video. For each experiment, we inpainted portions of the videos using both the sketch and text inputs as guides.

As presented in Table 4, MotionAura-L outperforms all other methods. The superior performance of MotionAura-L can be attributed to the combination of sketch and text inputs, which provide a richer context for guiding the inpainting process. While text descriptions offer a general understanding of the scene, the sketch provides explicit structural information, allowing the model to generate more accurate and coherent inpainting results. This dual-input method leads to better spatial alignment and temporal consistency, as evidenced by the higher scores when compared to MotionAura-L (w/o sketch), which relies solely on text input. The qualitative comparisons in Figure 10 show that MotionAura leads to more spatially and temporally coherent results than previous techniques.

Table 4: Quantitative evaluation for sketch-based inpainting. All models were pre-trained using the WebVid 10M -*recaptioned* and evaluated over the newly curated datasets comprising DAVIS and YouTube-VOS.

| Method | DAVIS (FVD↓ / CLIPSIM↑ / MQ↑ / TC↑) | YouTube - VOS (FVD / CLIPSIM / MQ / TC) |
|---|---|---|
| T2V (Khachatryan et al., 2023) | 782 / 0.2489 / 63.39 / 191 | 700 / 0.2601 / 64.28 / 193 |
| SimDA (Xing et al., 2023) | 752 / 0.2564 / 65.28 / 196 | 693 / 0.2699 / 67.82 / 195 |
| AnimateDiff (Guo et al., 2024) | 737 / 0.2401 / 68.97 / 199 | 685 / 0.2701 / 69.78 / 197 |
| CogVideoX-5B (Yang et al., 2024b) | 718 / 0.2512 / 73.37 / 205 | 677 / 0.2602 / 75.51 / 203 |
| MotionAura-L (w/o fft) | 689 / 0.2919 / 73.44 / 203 | 663 / 0.3164 / 75.87 / 202 |
| MotionAura-L (w/o sketch) | 687 / 0.3011 / 73.31 / 200 | 666 / 0.3061 / 75.14 / 199 |
| MotionAura-S | 704 / 0.2776 / 70.29 / 200 | 670 / 0.2892 / 71.28 / 199 |
| MotionAura-M | 692 / 0.2901 / 72.37 / 202 | 666 / 0.3119 / 73.47 / 200 |
| MotionAura-L | **685 / 0.3101 / 74.59 / 207** | **657 / 0.3511 / 76.62 / 204** |

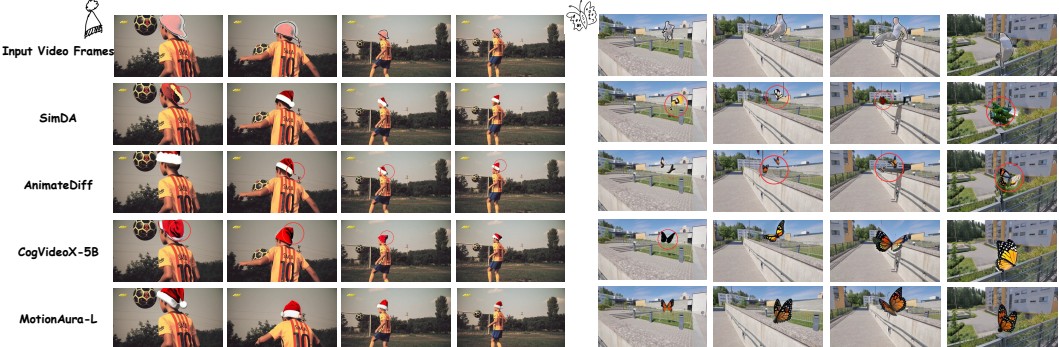

Figure 10: Sketch-guided inpainting results of various techniques. The red circles highlight the mistakes and misalignments, e.g., SimDA incorrectly inpaints the parts of the hat and the butterfly. AnimateDiff inpaints 2 butterflies in some frames, and inpaints half-transparent butterfly in the last frame. CogVideoX produces a hat with incorrect shape, and a black-color butterfly. By contrast, MotionAura produces high-quality and temporally consistent results.

## 5.4 ABLATION STUDY

**1.** We pretrain MotionAura on the WebVid 10M - recaptioned dataset and evaluate it on the MSR-VTT dataset for the text-conditioned video generation task. We evaluate the effect of the length of text description on the quality of the generated images. Increasingly detailed text descriptions provide better guidance and context. By virtue of this, the model captures finer nuances in scene composition, motion, and object interactions. Thus, the generated videos show superior quality and better match the user's requirements. Hence, with increasingly detailed descriptions, FVD reduces and CLIPSIM increases (Table 5).

Table 5: Impact of Text Length

| Length of Text | FVD | CLIPSIM |
|---|---|---|
| 30 | 364 | 0.2544 |
| 60 | 359 | 0.2598 |
| 90 | 355 | 0.2626 |
| 120 | 352 | 0.2654 |
| 150 | 349 | 0.2701 |
| 180 | 344 | 0.2823 |

Table 6: Impact of LoRA Rank

| LoRA Rank | FVD | CLIPSIM |
|---|---|---|
| 8 | 662 | 0.3291 |
| 16 | 660 | 0.3349 |
| 32 | 659 | 0.3421 |
| 64 | 657 | 0.3511 |

**2.** We evaluate the effect of rank of LoRA adaptors on the quality of sketch-based inpainting results on the YouTube-VOS dataset. A higher rank of the LoRA Adaptor involves updating more trainable parameters, which helps the model learn better representation and enhances model accuracy. Hence, a higher rank improves model performance, as shown in Table 6.

**3.** We evaluate MotionAura using sinusoidal embedding for the text-conditioned video generation. The FVD/CLIPSIM values are 379/0.2392 after pretraining on the WebVID 10M dataset and 349/0.2752 after pretraining on the WebVID 10M-recaptioned dataset. Clearly, RoPE embeddings (Table 3) provide superior results than the sinusoidal embeddings.

## 6 CONCLUSION

In this paper, we present *MotionAura*, a novel approach to generating high-quality videos given some text and sketch conditions. The videos generated by our model show high temporal consistency and video quality. MotionAura proposes several novelties, such as a new masked index loss during VAE pretraining, using FFT layers to segregate high-frequency features from the low-frequency ones and using RoPE embeddings to ensure better temporal consistency in denoised latents. Additionally, we recaption the WebVid-10M dataset with much more contextually rich captions, improving the denoising network's quality. Finally, we curate our datasets to address the task of sketch-guided video inpainting. Rigorous experiments show the effectiveness and robustness of *MotionAura* in video generation and subsequent downstream tasks.

**Acknowledgement:** Support for this work was provided by Science and Engineering Research Board (SERB) of India, under the project CRG/2022/003821. Jishu and Chirag contributed to this project while working as interns at IIT Roorkee.

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

# Supplementary Materials

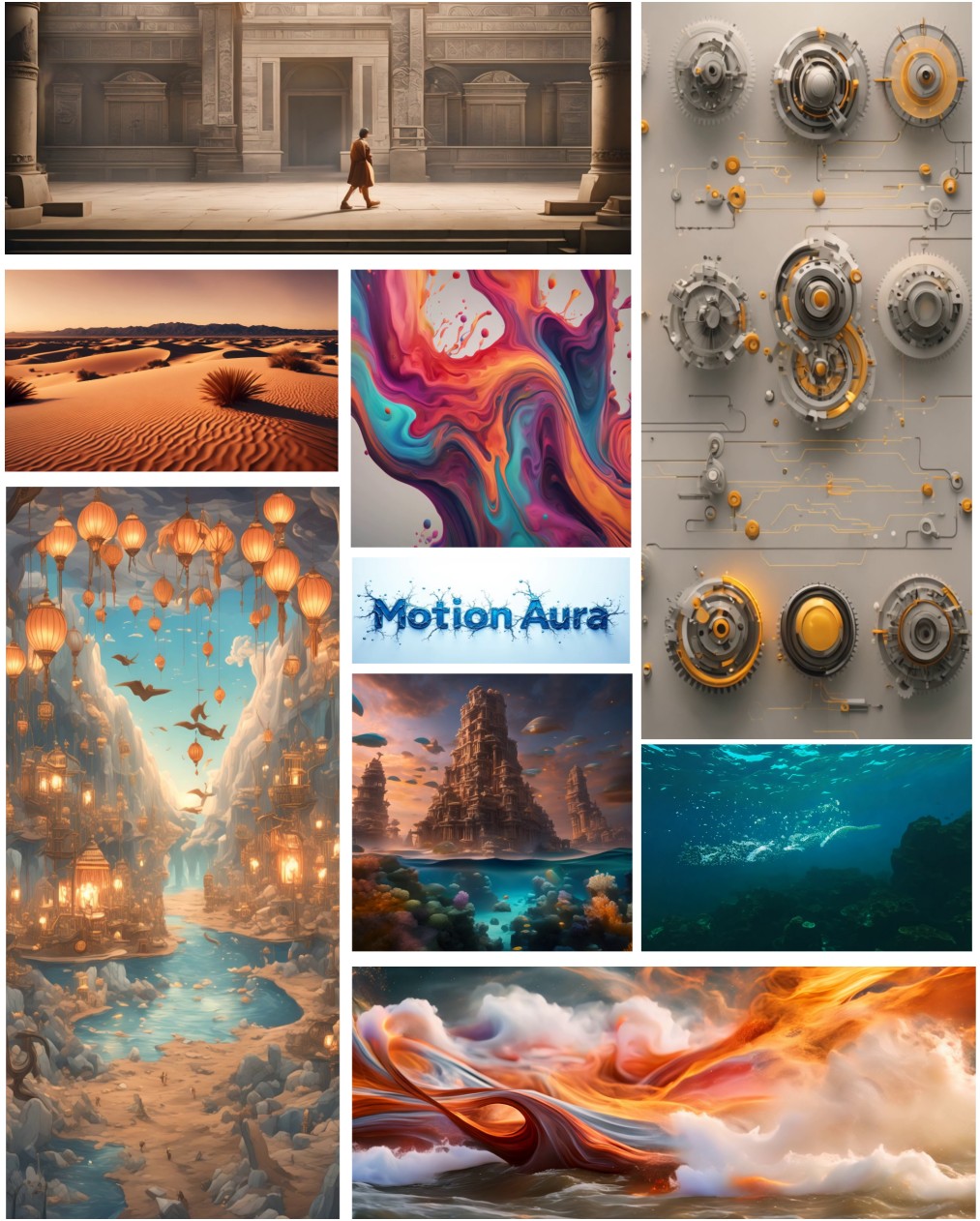

Figure S.1: Qualitative results for our proposed *MotionAura*. The corresponding videos can be found by clicking on the link.

## A 10 SECOND GENERATED SAMPLES

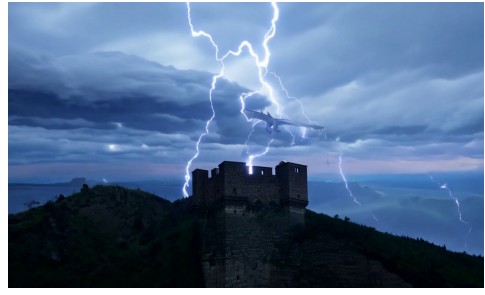

Figure S.2: Click here to see the video.

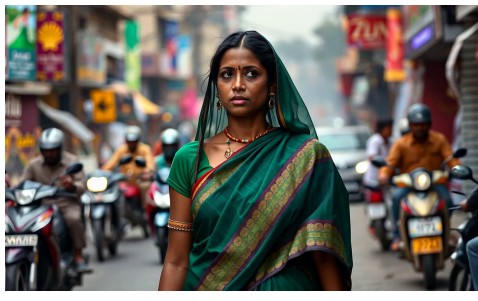

Figure S.3: Click here to see the video.

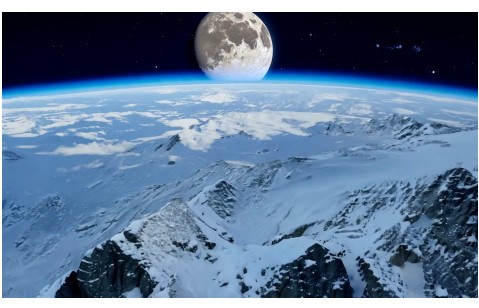

Figure S.4: Click here to see the video.

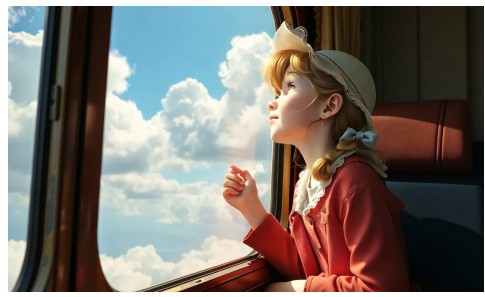

Figure S.5: Click here to see the video.

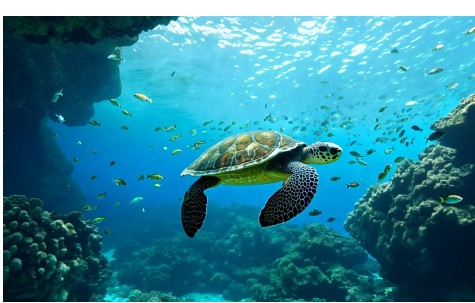

Figure S.6: Click here to see the video.

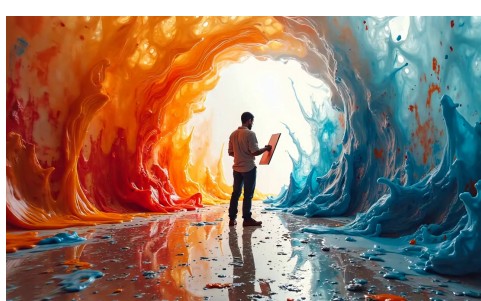

Figure S.7: Click here to see the video.

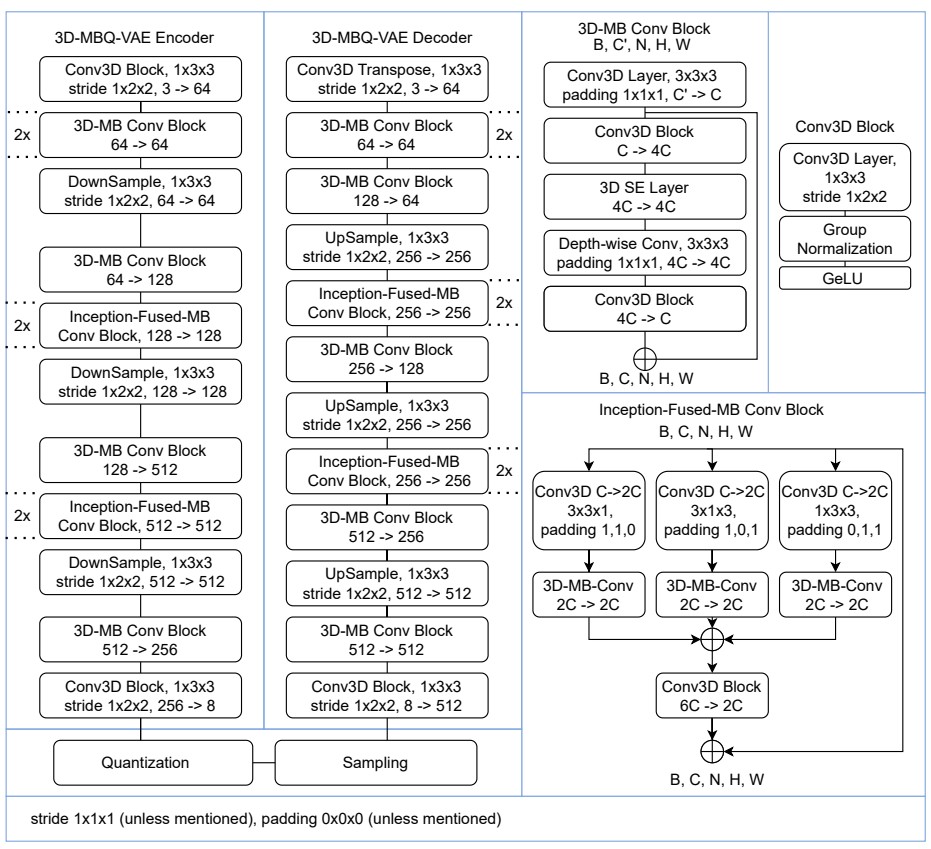

Figure S.8: 3D-MBQ-VAE architecture.

# B  3D-MBQ-VAE

## B.1  3D-MBQ-VAE ARCHITECTURE

Figure S.8 shows the 3D-MBQ-VAE (Mobile Inverted Vector Quantization Variational Autoencoder) architecture designed to efficiently encode and decode video data while utilizing both mobile inverted bottleneck blocks and vector quantization techniques. The MBQ-VAE framework is divided into two main parts: the encoder and the decoder, with quantization and sampling in between.

Encoder The encoder takes an input video with dimensions $B \times C \times N \times H \times W$, where $B$ is the batch size, $C$ is the number of channels, and $N \times H \times W$ represents the spatiotemporal dimensions (i.e., frames, height, and width). The encoding begins with a **Conv3D block** that captures spatiotemporal features using a 3D convolution with a kernel size of $1 \times 3 \times 3$, followed by a stride of $1 \times 2 \times 2$ to reduce spatial resolution while expanding the channel depth (e.g., from 3 to 64 channels). The encoder then applies multiple **3D-MB Conv blocks** (Mobile Inverted Conv Blocks) to capture complex features using depthwise separable convolutions, maintaining computational efficiency. These blocks are interspersed with **Inception-Fused-MB Conv blocks**, which improve feature extraction by combining multiple filter sizes and dimensions, enabling multi-scale processing.

At various stages, downsampling is performed via strided 3D convolutions to reduce the spatial and temporal dimensions further while increasing channel depth, helping the network focus on abstract, high-level features. This progression continues until the latent representation is compressed into a small-dimensional vector using a final **Conv3D block**. This compressed representation is passed to the **quantization layer**, where vector quantization is applied to discretize the latent variables for use in the VQ-VAE.

### B.1.1 DECODER

The decoder mirrors the encoder but in reverse, beginning with **vector sampling** from the quantized latent space. It starts with an initial **Conv3D block** to upsample the channels from the quantized latent vector. The network then proceeds through several **3D-MB Conv blocks** and **Inception-Fused-MB Conv blocks**, gradually increasing the spatial resolution via **Upsampling layers**. Combined with inverted convolution blocks, these layers allow the decoder to reconstruct fine-grained details from the compressed latent vector.

The upsampling process continues in steps, restoring the spatial and temporal dimensions while reducing the channel depth to match the input dimensions. The final layer of the decoder uses a **Conv3D Transpose** operation to transform the latent representation back to the original input resolution, ensuring the correct number of channels is maintained for video reconstruction.

### B.1.2 KEY FEATURES

- **Mobile Inverted Bottleneck Convolutions**: The use of 3D-Mobile Inverted Bottleneck (MB) Conv blocks enables the model to achieve high efficiency by expanding and contracting the number of channels at different stages while preserving critical features through depthwise separable convolutions.
- **Inception-Fused Convolutions**: By incorporating multiple kernel sizes in the Inception-Fused blocks, the architecture can process multi-scale spatial and temporal features, improving video compression and reconstruction quality.
- **Vector Quantization**: The quantization step allows for discretizing the latent space, which is critical for VQ-VAE models to perform learned compression, making this architecture suitable for generative tasks and compression with high efficiency.

MBQ-VAE is an efficient and scalable video processing architecture, leveraging mobile inverted convolutional blocks and vector quantization to balance computational efficiency with high-quality reconstruction of video data. This makes it well-suited for video compression, generation, and reconstruction tasks.

### B.1.3 MOBILE-INVERTED 3D CONV BLOCK

The 3D Mobile Inverted Convolution (3D-MB Conv) Block plays a crucial role in our VQ-VAE network by enabling efficient spatiotemporal feature extraction with reduced computational complexity. This block begins with a 3D convolutional layer that maps from a higher dimensional input space $C'$ to the desired channel dimension $C$, utilizing a $3 \times 3 \times 3$ kernel with padding to maintain spatial and temporal resolution. The feature maps are then expanded using a pointwise 3D convolutional block from $C$ to $4C$, enhancing the feature representation without significantly increasing computation. A 3D Squeeze-and-Excitation (SE) layer follows, which adaptively recalibrates the channel importance, improving the model's sensitivity to informative spatiotemporal features. Next, a depthwise separable 3D convolution applies a $3 \times 3 \times 3$ kernel to each feature map independently, maintaining $4C$ channels and effectively reducing computation compared to standard convolutions. Finally, a 3D convolutional block reduces the channel dimension back from $4C$ to $C$, ensuring that the block output is consistent with the input dimensionality. With its inverted residual connections and depthwise convolutions, this overall structure enhances efficiency in capturing hierarchical spatiotemporal patterns, making it highly suitable for the reconstruction and compression tasks in our VQ-VAE network.

### B.1.4 INCEPTION-FUSED MB CONV BLOCK

The 3D Inception-Fused-MB Conv Block is a sophisticated convolutional module designed to efficiently capture both spatial and temporal features in video data, commonly used in tasks like Video VQ-VAE (Vector Quantized Variational Autoencoder) shown in Figure S.8. This block employs an Inception-style structure with three parallel 3D convolution paths, each with different kernel configurations (e.g., $3 \times 3 \times 1$, $3 \times 1 \times 3$, and $1 \times 3 \times 3$) to capture multi-scale spatiotemporal patterns in various dimensions. Each branch is followed by a depthwise separable 3D-MobileBlock (MB-Conv) that reduces computational complexity while maintaining efficiency. The outputs of these paths are concatenated, allowing the network to fuse multi-scale features, which are then processed

by a final 3D convolutional block. This design enables efficient handling of complex video representations in VQ-VAEs, optimizing the model's ability to compress and reconstruct video sequences while preserving key spatial and temporal information. We perform temporal downsampling within the 3D-MBQ-VAE encoder at specific stages. Temporal downsampling is applied by a factor of 2 at the 6th block (Downsampling Block) and again by a factor of 2 at the 9th block, resulting in a total downsampling factor of 4 along the temporal dimension.

## B.2 Additional Abblation Studies for 3D-MBQ VAE

### B.2.1 Impact of varying codebook sizes on video reconstruction

We conducted a comprehensive ablation study analyzing the effects of different codebook sizes, embedding dimensions, and quantization techniques on video reconstruction quality. In our default configuration, we employ the Lookup-Free Quantization (LFQ) method from MAGVIT-v2, utilizing an embedding size of $d = 8$ and a codebook vocabulary size of $C = 12,800$.

Table 7: Impact of codebook size ($C$) on video reconstruction (d= Embedding Size)

| Method Of Quantization | $C$ | $d$ | Time (ms) | COCO-Val PSNR/ SSIM/ LPIPS | WebVid-Val PSNR/ SSIM/ LPIPS |
|---|---|---|---|---|---|
| VQ | 256 | 512 | 35 | 29.96/ 82.21/ 0.133 | 30.01/ 81.89/ 0.1441 |
| Gumbal Quantization | 256 | 512 | 23 | 29.83/ 81.98/ 0.140 | 29.78/ 80.56/ 0.1511 |
| VQ | 1024 | 256 | 47 | 30.26/ 82.54/ 0.130 | 30.72/ 81.09/ 0.1453 |
| VQ | 4096 | 256 | 56 | 30.72/ 83.11/ 0.127 | 31.03/ 82.22/ 0.1331 |
| VQ | 8000 | 128 | 69 | 30.45/ 82.19/ 0.144 | 30.98/ 83.36/ 0.1299 |
| LFQ | 8000 | 16 | 20 | 31.08/ 84.34/ 0.112 | 31.98/ 85.04/ 0.1103 |
| LFQ(Default) | 12800 | 8 | 22 | 31.22/ 84.78/ 0.092 | 32.09/ 85.78/ 0.1081 |

The primary reasons for adopting LFQ over traditional Vector Quantization (VQ) methods are:

**1. Larger Vocabulary with Smaller Embedding Size:** LFQ enables the use of a considerably larger codebook vocabulary while maintaining a compact embedding size. This approach enhances the model's expressive power without significantly increasing computational costs. By leveraging a larger vocabulary with smaller embeddings, the model benefits from higher codebook utilization, a strategy that has been shown to improve performance Sun et al. (2024c) for the task of image reconstruction.

**2. Improved Efficiency:** LFQ is faster and more optimized than standard VQ methods, which is beneficial for processing high-resolution video data.

### B.2.2 Results with random masking and full-frame masking

For random spatial masking, we employ a "cosine scheduling" strategy, progressively varying the masking ratio from 20% to 60% during training. This scheduling helps the model gradually adapt to different levels of partial observations, thereby enhancing robustness in spatial feature extraction.

For frame masking, we similarly utilize cosine scheduling, adjusting the masking ratio from 10% to 50%. This approach ensures a balanced training dynamic where the model is exposed to varying levels of temporal information occlusion, ultimately aiding in better temporal coherence and reconstruction quality. These masking strategies are designed to incrementally challenge the model, helping it learn to reconstruct meaningful content under different masking conditions and thus improving overall generalization.

Table 8 shows the results with random masking and full-frame masking. These results underscore the significance of both random masking and fully masked frames in enhancing the model's ability to predict frame indices effectively.

The random masking strategy compels the model to infer missing spatial information from partial observations within each frame, effectively enhancing its ability to capture spatial dependencies and structures. By being trained on randomly masked data, the model learns to reconstruct or predict spatial details based on contextual cues, leading to stronger spatial representations. Conversely,

Table 8: Ablation results with random masking and full-frame masking

| | COCO-Val | WebVid-Val |
|---|---|---|
| Methods | PSNR/ SSIM/ LPIPS | PSNR/ SSIM/ LPIPS |
| W/o Random Masking | 29.17/ 81.92/ 0.111 | 28.78/ 82.29/ 0.1334 |
| W/o Full Frame Masking | 30.09/ 82.22/ 0.102 | 30.11/ 83.01/ 0.1404 |
| W/o Random Masing and Full frame Masking | 28.82/ 80.77/ 0.124 | 26.77/ 79.92/ 0.1552 |
| With Random Masing and Full frame Masking | 31.22/ 84.78/ 0.092 | 32.09/ 85.78/ 0.1081 |

supervision for fully masked frame index prediction requires the model to determine the correct temporal order of completely obscured frames, which enhances temporal consistency. This task forces the model to understand and model temporal dynamics across frames, as it must rely on learned temporal patterns to accurately predict the positions of fully masked frames within the sequence.

### B.2.3 VIDEO COMPRESSION RESULTS ON MCL-JCV DATASET

We conducted a zero-shot inference on 30 videos from the MCL-JCV dataset (Wang et al. (2016)), resized to a resolution of 640 x 360, which aligns with the experimental setup used by MAGVIT Yu et al. (2023a) and MAGVIT-v2Yu et al. (2023c) . We evaluate compression quality using standard distortion metrics (LPIPS, PSNR, and SSIM) at a bit rate of 0.0384 bpp (bits per pixel). The results are shown in Table 9. At equivalent bit rates, traditional codecs such as HEVC Sullivan et al. (2012) and VCC Bross et al. (2021) may sometimes achieve finer local detail rendering compared to 3D-MBQ-VAE. However, they often introduce block artifacts that, while detrimental to perceptual quality, are not adequately captured by PSNR and SSIM. This is reflected in the LPIPS metric.

Table 9: Video compression results on MCL-JCV dataset

| Methods | PSNR | SSIM | LPIPS |
|---|---|---|---|
| HEVC | 30.1 | 0.943 | 0.199 |
| VCC | **32.65** | **0.966** | 0.153 |
| MAGVIT | 23.7 | 0.846 | 0.144 |
| MAGVIT-v2 | 26.18 | 0.894 | 0.104 |
| 3D-MBQ-VAE (Ours) | 29.09 | 0.922 | **0.089** |

## C  DATASETS

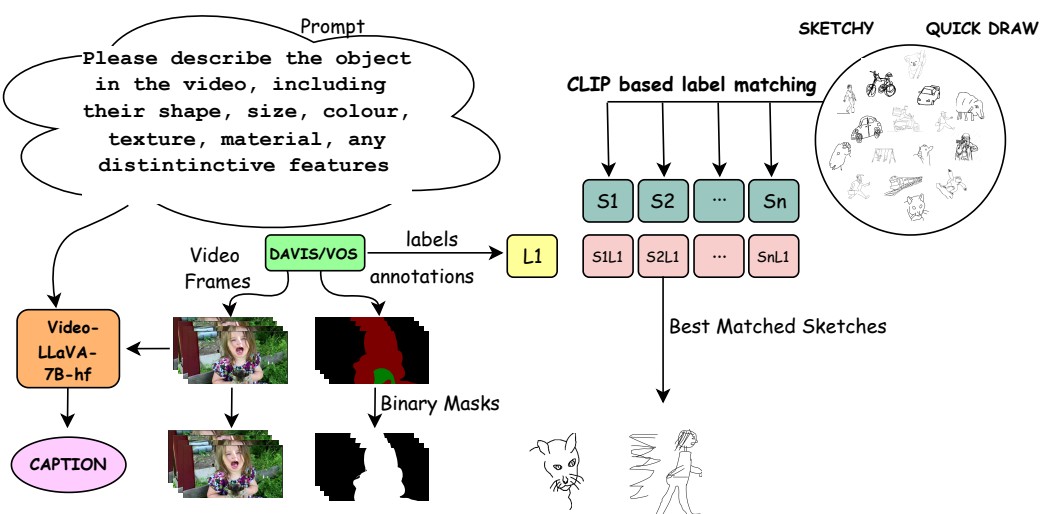

Figure S.9: For dataset curation for the downstream task of sketch-guided video inpainting, captions have been generated using Video-LLaVA-7B, and the corresponding sketches were obtained using CLIP-based label matching

The following section discusses the datasets used for pretraining and training our proposed models. The dataset selection was based on the complexity and nature of each task. We also discuss the process of caption regeneration for the WebVid-10M dataset and the curation of a new dataset for the task of Sketch-based Video Inpainting.

## C.1 DATASETS USED FOR 3D MBQ-VAE PRE-TRAINING

For pre-training our 3D MBQ-VAE, we use the YouTube 100M Hershey et al. (2017) dataset. The large number of videos and the comprehensive pretraining help our 3D-MBQ-VAE learn an efficient latent representation of video frames in a reduced spatiotemporal domain.

## C.2 T2V MODEL TRAINING

For training the diffusion model, we use the Webvid-10M Bain et al. (2022) with text as the condition. We find that the original text captions in the Webvid-10M dataset do not provide adequate guidance for diffusion pretraining purposes (Table 3). Hence, we re-caption the entire dataset with much more detailed prompts for every video. The corresponding text pairs given as conditions for diffusion pretraining are generated by LLaVA Next- 34B Li et al. (2024). In both cases, we had an 80-20 split between the training and test set. The following prompt was given to the LLM for regenerating the textual description of the video:

---

**Prompt**

```
Explain in detail about the given video with:
    • Focus on the primary subject and any important actions
      or interactions.
    • Highlight key details, such as distinctive features,
      expressions, or objects.
    • Describe the setting and environment, noting any
      relevant background elements.
    • Convey the mood or atmosphere of the scene.
    • Combine these elements into a clear, concise caption
      that accurately represents the image's content and
      context.
```

---

## C.3 PREPROCESSING TO DEAL WITH THE WATERMARKS IN THE WEBVID10M DATASET

To address the issue of watermarks in the WebVid-10M dataset and ensure that our generated videos do not exhibit these artifacts, we implemented a preprocessing strategy involving algorithmic inpainting methods. Specifically, we identified and masked the regions in each video frame that contained watermarks. We then applied inpainting techniques to these masked areas to reconstruct the underlying content, effectively removing the watermarks while preserving the overall visual quality of the videos. Although this process may introduce minor visual artifacts, it prevents the model from learning and replicating watermark patterns during training. Consequently, our model is able to generate clean, watermark-free videos.

Furthermore, to enhance the resolution and temporal consistency of the preprocessed videos, we employed the StableVSR Rota et al. (2024) video super-resolution model. This step refines visual details and mitigates any reduction in quality resulting from the inpainting process, ensuring that the dataset used for training is of high quality and temporally consistent.

## C.4 TRAINING FOR SKETCH GUIDED VIDEO INPAINTING TASK

For the downstream task of Sketch Guided Video Inpainting, we required a dataset containing the videos, text prompts, binary masks, and corresponding object sketches. To our knowledge, no such open-source dataset satisfies all our requirements. Hence, we curated our own dataset to fit our requirements. Figure S.9 shows the entire process of how we curate the dataset for the downstream

task using two datasets primarily used in the video inpainting research community: Youtube-VOS Xu et al. (2018) and DAVIS Perazzi et al. (2016).

# D    IMPLEMENTATION DETAILS

We now present the implementation details and parameters used in our study.

## D.1    HYPERPARAMETERS OF VARIOUS TECHNIQUES

To ensure a fair comparison, we utilized the respective hyperparameters recommended in the original papers for each method. Table 10 outlines the specific hyperparameters used for training and inference across all baseline methods and our proposed approach.

Table 10: Hyperparameters of various techniques

| Training H-Parameters | SimDA | AnimateDiff | CogVideoX-5B | MotionAura |
|---|---|---|---|---|
| Learning Rate | 1.00E-05 | 1.00E-04 | 1.00E-04 | 1.00E-05 |
| Gradient Accumalation Steps | 4 | 4 | 8 | 8 |
| Batch Size Per GPU | 8 | 8 | 2 | 3 |
| Optimizer | AdamW | AdamW | AdamW | AdamW |
| Lr-Schedular | Linear | Cosine | Cosine | Cosine |
| Epochs | 35 | 30 | 40 | 30 |
| Noise Schedular | DDPM | DDPM | FlowMatching | VQDiffusionScheduler |
| Diffusion Steps | 500 | 500 | 100 | 30 |
| Training Precision | Float16 | Float16 | BFloat16 | BFloat16 |
| GPUs | 4 x 8 A100 | 4 x 8 A100 | 8 x 8 A100 | 6 x 8 A100 |
| Text Encoders | T5-XL | CLIP | T5-XXL | T5-XXL |
| Time Embedding Size | 256 | 256 | 512 | 512 |
| Gradient Clipping | 1 | 1.5 | 2.5 | 2.5 |
| Max Text Length | 77 | 128 | 200 | 256 |
| Embedding Size | 1024 | 1024 | 4096 | 4096 |
| CFG Scale | 8.5 | 8 | 10.5 | 10 |
| Positional Encodings | Sinusoidal | Sinusoidal | RoPE | RoPE |

## D.2    IMPLEMENTATION DETAILS OF 3D-MBQ-VAE PRE-TRAINING

We train our 3D MB-VAE model on the YouTube100M video dataset using four different configurations of frames and resolutions. These configurations are: (1) 32 frames at $256 \times 256$ resolution, (2) 16 frames at $512 \times 512$ resolution, (3) 32 frames at $1280 \times 720$ resolution, and (4) 48 frames at $640 \times 480$ resolution. The corresponding batch sizes for these configurations are 8, 4, 2, and 4, with sampling ratios of 40%, 10%, 25%, and 25%, respectively. We apply gradient accumulation steps of 8 across all configurations.

The AdamW optimizer is employed with a base learning rate of $1 \times 10^{-4}$ with cosine learning rate decay. To reduce the risk of numerical overflow, we train the 3D MB-VAE model in `float32` precision. The training is performed across 4 nodes, each equipped with 8 NVIDIA A100 GPUs (80 GB each), for a total of 800,000 training steps.

## D.3    IMPLEMENTATION DETAILS OF DIFFUSION PRETRAINING

We train our denoiser on the Webvid-10M dataset for diffusion model pretraining. The AdamW optimizer uses a learning rate of $1 \times 10^{-5}$ with linear learning rate decay. During training, we

explore multiple frame settings: 16, 32, 48, and 64 frames. For 16-frame training, we use a batch size of 512 and train for 120,000 iterations. For 32, 48, and 64 frames, the batch sizes are 256, 256, and 128, respectively, with corresponding training durations of 120,000 iterations for 32 frames, 150,000 iterations for 48 frames, and 190,000 iterations for 64 frames. This training is conducted on 8 nodes, each equipped with 8 NVIDIA A100 GPUs (80 GB memory per GPU). We utilize dynamic resolution and aspect ratio adjustments during training to enhance the model's robustness to varying input sizes.

### D.4    IMPLEMENTATION DETAILS OF SKETCH-BASED INPAINTING TRAINING

For the downstream tasks, we fine-tune the model using LoRA (Low-Rank Adaptation) parameters specifically for the K and Q projection weights. The training is carried out with a batch size of 8 per GPU at a resolution of $256 \times 512$. This stage is performed on 2 nodes, each with 8 NVIDIA A100 GPUs (80 GB memory per GPU). The training includes 2,000 iterations for the VOS dataset and 2,600 iterations for the DAVIS dataset. We employ the Adam optimizer with an initial learning rate of $2 \times 10^{-4}$, following a cosine decay schedule. Gradient accumulation steps are set to 8 to manage memory and computational load, enabling efficient training through mixed-precision techniques.

### D.5    SETTINGS FOR JOINT TRAINING WITH VIDEO AND IMAGES

For joint training with video and images, we used the JDB Sun et al. (2024a) dataset, which is a large-scale image dataset featuring around 4 million high-resolution images from Midjourney. We set $N \geq 16$ for videos or $N = 1$ for images to have a setup akin to W.A.L.T Gupta et al. (2025).

### D.6    USE OF RoPE VS SINUSOIDAL POSITIONAL EMBEDDINGS

The use of RoPE (Rotary Positional Embeddings) in the attention layers of the denoising network shows a faster convergence, as seen in Figure S.10. RoPE (Rotary Positional Embeddings) is a type of positional encoding designed for transformer models to encode relative position information flexibly. RoPE embeddings support larger context lengths, enabling the model to generalize better over varying input sizes. Since they capture positional relationships based on rotation, the transformer can effectively handle and reason over sequences of greater lengths. This is handy when dealing with high-dimensional video frames, thus helping the model learn effectively.

## E    COMPARATIVE RESULTS WITH DISCRETE AND CONTINUOUS SPACE

We now present our motivation for transitioning the diffusion model paradigm from continuous space to discrete space.

**1. Better Representation through Discrete Quantization:** By moving to a discrete latent space, we are able to leverage powerful vector quantization approaches, which help in capturing the essential features of high-dimensional data more efficiently. The discrete representation obtained from quantization often yields better compression and a more structured representation of complex data, which helps the diffusion process to operate over a more compact and semantically meaningful latent space. This can lead to improved training stability and generation quality, especially for high-dimensional domains such as video.

**2. Reduction in Computational Complexity:** Traditional diffusion models applied in continuous space can require substantial computation, particularly during inference, as they operate directly on high-dimensional latent representations. By quantizing the data into discrete tokens, we can reduce the dimensionality and the complexity of the diffusion process. This makes the inference more efficient without compromising the model's expressiveness. Recent works, such as MAGVIT Yu et al. (2023a), have shown that combining discrete latent space with the diffusion framework can yield efficient and high-quality results for complex generative tasks, such as video synthesis.

**3. Alignment with Discrete Decoders:** Utilizing a discrete latent space allows us to align more effectively with certain discrete decoder architectures, such as transformers, which operate over token sequences. This leads to improved synergy between the encoder-decoder framework, where the encoder maps the input data into discrete tokens and the decoder operates over those tokens

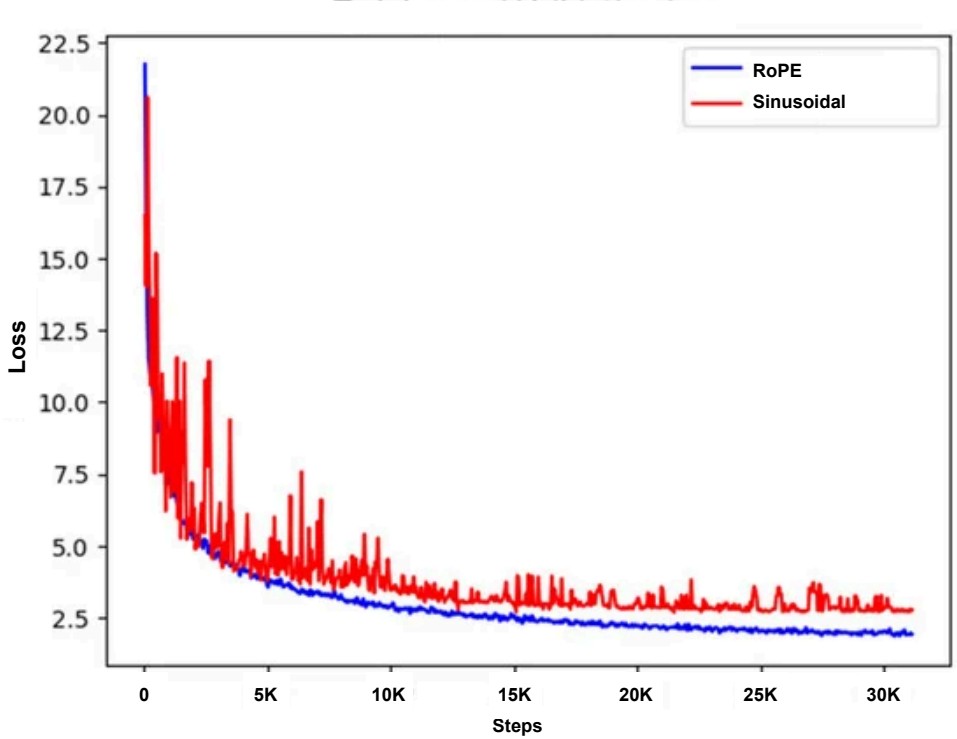

Figure S.10: RoPE embeddings show faster convergence demonstrating superior learning compared to Sinosoidal embeddings

to generate the output. The performance gains we observed were not solely due to the encoder being more performant in this setting, but also due to the effective interaction between discrete representation and token-based decoders.

**4. Enhanced Control and Semantic Richness:** Discrete tokens are more interpretable and often exhibit greater semantic richness, which can be beneficial for downstream generative tasks where controllability and interpretability are desirable. This is particularly advantageous for tasks like text-to-video generation or inpainting based methods., where mapping high-level semantics to the generative model plays a crucial role. Recent works, such as Palette Saharia et al. (2022), VQ-Diffusion Gu et al. (2022), and Sketch-based Video Inpainting Sharma et al. (2024), have successfully employed discrete latent spaces, demonstrating the effectiveness of this approach across a variety of generative tasks.

Note that $L_{\mathrm{mfi}}$ is used to compute the probability of the index corresponding to a completely masked frame. It functions as a standard discriminative loss, specifically equivalent to $-\log(P_\theta(V_i))$, where $V_i$ represents the masked frame. This loss term is crucial for ensuring that the model can accurately predict and reconstruct the masked frames by providing a probabilistic grounding.

Table 11 compares the results of discrete space with continuous space. Notice that the results of discrete space are superior. Our findings further indicate a significant improvement when $L_{\mathrm{mfi}}$ loss is applied in continuous latent space, demonstrating its effectiveness in enhancing the model's performance.

Table 11: Comparison between discrete and continuous spaces

|  | COCO-Val | WebVid-Val |
| --- | --- | --- |
| Methods | PSNR/ SSIM/ LPIPS | PSNR/ SSIM/ LPIPS |
| Discrete space | 31.22/ 84.78/ 0.092 | 32.09/ 85.78/ 0.108 |
| Continous Space (w/o $L_{\text{mfi}}$) | 28.27/ 82.29/ 0.155 | 29.01/ 80.45/ 0.166 |
| Continous Space | 31.09/ 83.96/ 0.109 | 31.62/ 85.05/ 0.112 |

# F  SKETCH-GUIDED VIDEO GENERATION

## F.1  CONDITION INJECTION FRAMEWORK

Given a masked video, the primary objective of our approach is to reconstruct the unmasked frames using the surrounding video context, guided by sketch conditions. The inpainting process begins by encoding the masked video into a compact latent space. This encoding step ensures the efficient capture of essential information about the video, facilitating robust modeling of temporal and spatial relationships. Simultaneously, the sketch conditioning is encoded into a complementary latent space, enabling the framework to effectively integrate external guidance.

During fine-tuning, the model is trained with a combination of the masked video, its unmasked counterpart, and the corresponding sketch conditions. This multi-input paradigm allows the model to learn the nuances of filling masked regions in alignment with both the video context and the provided sketches. Our hypothesis is that incorporating external conditioning improves the model's ability to interpret masked regions, enabling precise and context-aware attention over these regions. This approach aims to refine the synthesis of missing areas while adhering to the desired object shapes and poses specified by the sketches.

To address the potential challenge of catastrophic forgetting during continual or lifelong learning, we integrate a Low-Rank Adaptation (LoRA) module into the fine-tuning process. LoRA adapters are introduced as lightweight parameter-efficient layers, allowing the model to adapt to new tasks without full end-to-end fine-tuning. This decision is grounded in the need for scalability and the prevention of knowledge erosion from previously learned tasks. By incorporating LoRA adapters, we aim to achieve adaptability comparable to modern language models while maintaining robustness in learned representations.

The placement of LoRA adapters is determined using Elastic Weight Consolidation (EWC), a technique that identifies model layers with higher activation gradients and step gradients. By targeting these layers, the adapter network maximizes its effectiveness, ensuring that the critical components of the model's architecture are adapted with precision and efficiency.

During inference, the model utilizes the masked video and sketch conditioning as inputs. The encoded representations are processed by a generative network to predict and reconstruct missing regions. The generative network ensures temporal consistency across frames while aligning the synthesized content with the sketches, resulting in video outputs that maintain fidelity to the original content and the provided guidance.

In summary, while the condition injection framework may appear straightforward, our design incorporates several nuanced techniques, including efficient latent space encoding, fine-tuning with LoRA adapters, and strategic application of EWC-based insights. These innovations collectively enhance the model's ability to perform sketch-guided video inpainting effectively and adaptively.

## F.2  USE OF LoRA ADAPTORS FOR PEFT

LoRA adaptors are beneficial in Parameter-Efficient Fine-Tuning (PEFT) because they significantly reduce the number of parameters that must be trained while maintaining the model's performance. In large-scale models, fine-tuning all the parameters is computationally expensive. LoRA adaptors introduce trainable, low-rank matrices into the model layers, which allows only a small subset of parameters to be updated during training. This drastically reduces the memory and computational overhead required for fine-tuning, making it more feasible to adapt large models to new tasks using limited data and resources. By optimizing only a small fraction of the parameters, LoRA adaptors also help prevent overfitting and make the fine-tuning process more efficient without compromising

model accuracy. As our model is very large, thoroughly training it from scratch for every downstream task does not make sense. Hence, we use the LoRA adaptors in our Spectral Transformer to adapt to the downstream task. Fig S.11 shows the details of our Spectral Transformer with the LoRA adaptors. Two types of LoRA have been used as shown in Fig S.11. One is the Attention

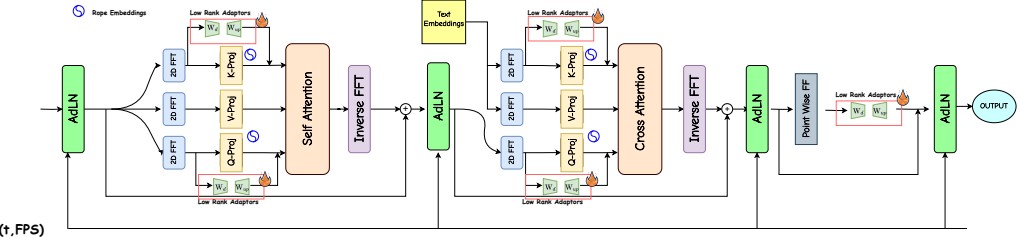

Figure S.11: Use of LoRA in the above helps to efficiently fine-tune the above spectral transformer for the downstream task while only using a fraction of the number of trainable parameters.

LoRA, which is applied in parallel to the Attention Blocks. This LoRA is applied only to the Key and the Query vectors. The reason for applying LoRA here is to ensure the base model retains information from previously trained tasks. LoRA is not applied to the Value vector due to Elastic Weight Compression. Elastic Weight Compression identifies the most critical parameters the model must remember to learn a task effectively. Using Elastic Weight Compression, we find that the parameters of the Key and Query are the most crucial when adapting to a new task, which is why we apply LoRA to them.

The second type of LoRA used is the Feed Forward LoRA, applied sequentially to the final Pointwise Feed Forward Layer. The Feed Forward Layer is a highly dense network, and if LoRA were used in parallel to this layer, followed by concatenation with its output, it would not result in meaningful representations. The sequential application ensures that the model learns effective and compact representations.

## F.3 SAMPLE SKETCH-GUIDED VIDEO GENERATION RESULTS

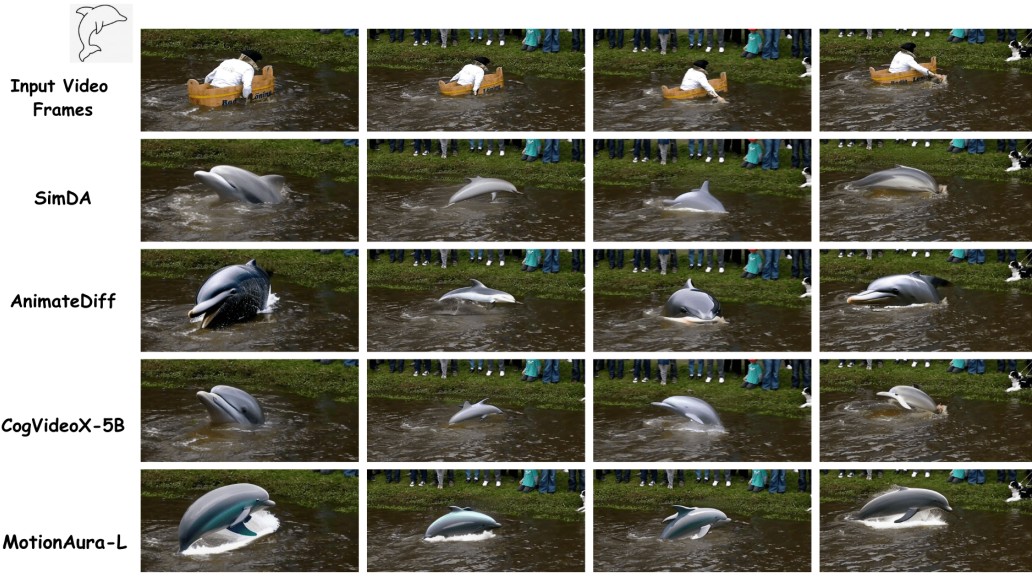

Figure S.12: Sketch-guided video generation results for a dolphin sketch

Figures S.12 and S.13 show sample sketch-guided video generation results of various techniques. Given the inpainting video results, it is not hard to see that our model achieves the best results in terms of temporal consistency and preserving the inpainted object throughout the frames.

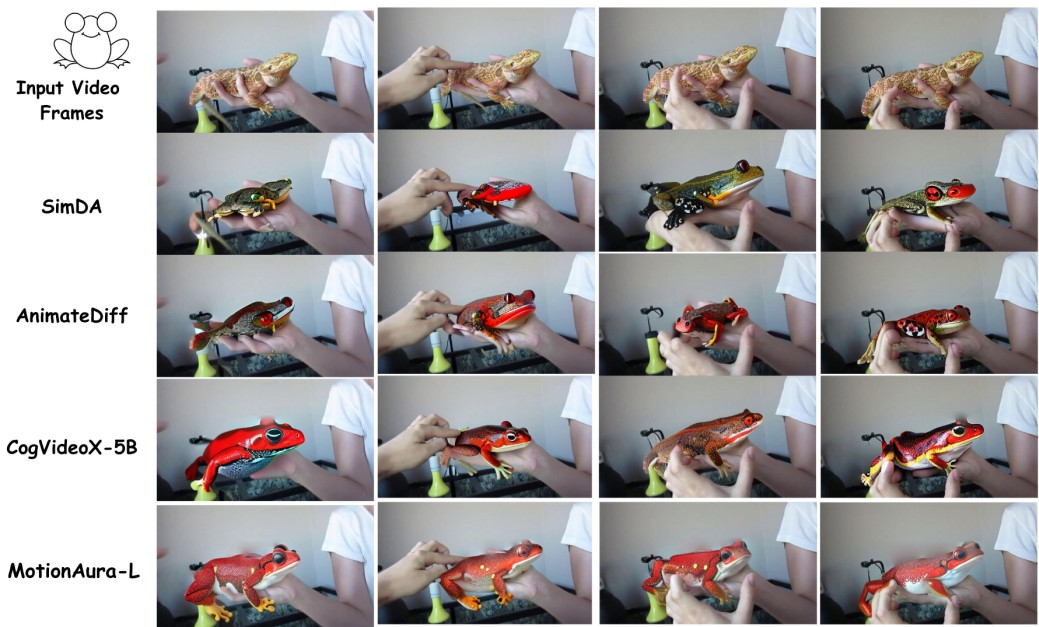

Figure S.13: Sketch-guided video generation results for a frog sketch

## G  TEXT-GUIDED VIDEO GENERATION

### G.1  ZERO-SHOT RESULTS ON UCF-101 DATASET

To highlight the generalization capabilities of our model across diverse datasets, we present its performance on the UCF-101 dataset, as summarized in Table 12. The results clearly demonstrate the superiority of MotionAura compared to prior works. Specifically, the table showcases the performance of various methods using different schedulers during inference, paired with their respective preferred CFG (Classifier-Free Guidance) scales.

Notably, MotionAura-L achieves impressive results while requiring only 10 steps, making it significantly faster than other baseline methods. This efficiency underscores the practical advantages of MotionAura-L in scenarios demanding high-speed inference without compromising performance.

Table 12: Zero-shot results on UCF-101 dataset

| Methods | Scheduler | CFG Scale | Steps | FVD |
|---|---|---|---|---|
| SimDA | EulerAncestralDiscreteScheduler | 8.5 | 30 | 300 |
| AnimateDiff | DPMSolverMultistepScheduler | 8.0 | 30 | 277 |
| CogVideoX-5B | EulerAncestralDiscreteScheduler | 10.5 | 25 | 239 |
| MotionAura-L | VQDiffusionScheduler | 8.5 | 10 | 219 |

### G.2  SAMPLE TEXT-GUIDED VIDEO GENERATION RESULTS

Figures S.1 to S.7, and Figures S.14 to S.20 further demonstrate the robustness of our model on different tasks.

**Abstract Sculptures**

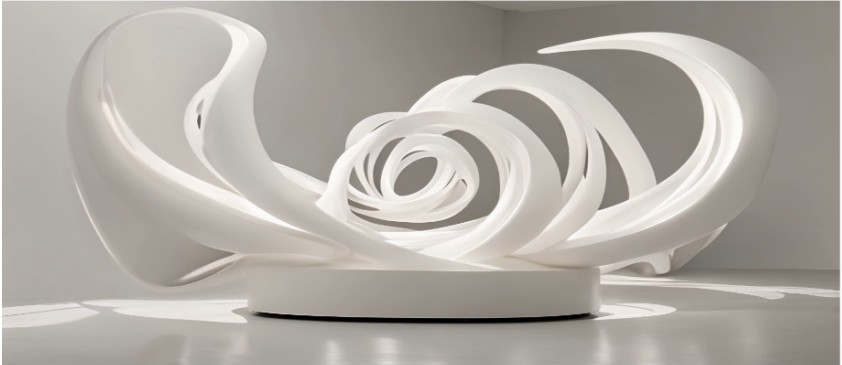

*Create an abstract video featuring fluid, metallic sculptures morphing and twisting in a serene, white space. The camera circles around the forms, capturing their smooth surfaces as they shift between shapes. Reflections play across their surfaces, creating a mesmerizing, almost hypnotic effect.*

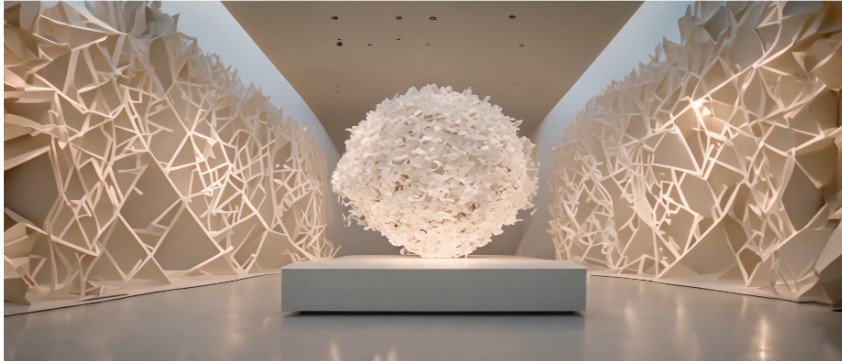

*Render is a video showcasing abstract sculptures made of geometric shapes fused with organic elements. The camera moves through a gallery-like space, focusing on each sculpture's intricate details. The forms appear to grow and change, as if they are alive, with soft lighting emphasizing their textures.*

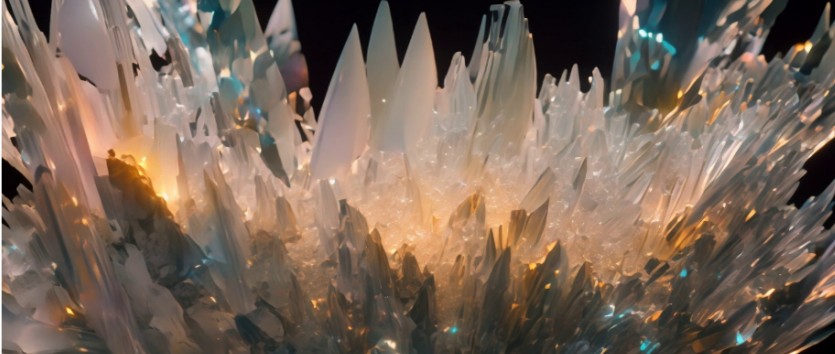

*Generate a video featuring abstract sculptures that resemble intricate crystal formations. The camera pans slowly, revealing the sharp angles and translucent surfaces of the sculptures. The light refracts through the crystals, creating a dazzling display of colors and reflections against a dark background.*

Figure S.14: Abstract sculptures (Click here to see the video.)

**Cinematic Videos**

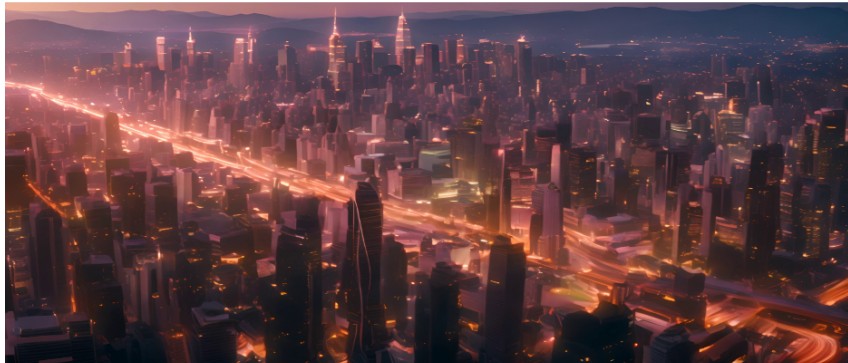

*Create a cinematic video featuring a sprawling, futuristic city at dusk. Skyscrapers tower above, casting long shadows. The camera slowly zooms out, revealing the bustling city streets below, filled with neon lights and fast-moving traffic. A heavy atmosphere of anticipation fills the air as dark clouds gather on the horizon.*

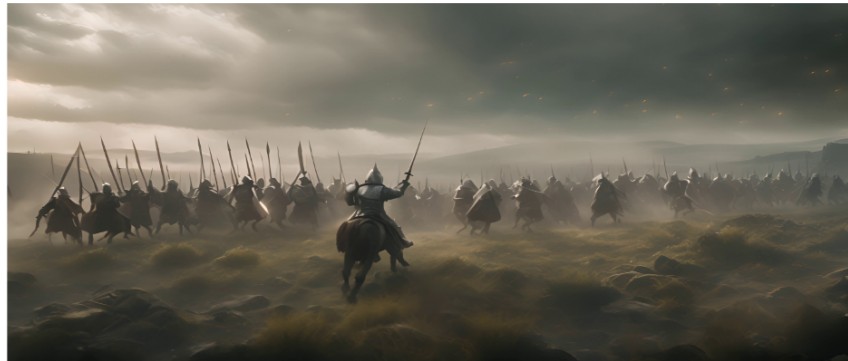

*Render is a cinematic video of an epic battle in a medieval fantasy world. Armored knights clash with mythical creatures on a foggy battlefield. The camera captures intense close-ups of swords clashing and panoramic views of the chaotic scene. The sky is overcast, with the sound of thunder rumbling in the distance.*

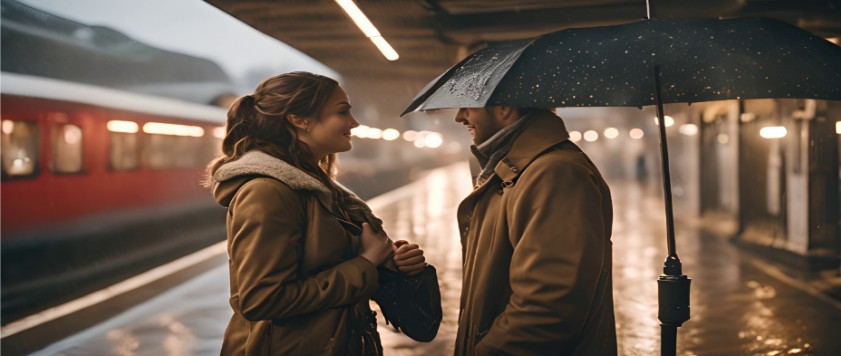

*Create a cinematic video of a couple reuniting at a train station in the pouring rain. The camera focuses on their expressions as they embrace, with raindrops glistening on their faces. Soft, warm lighting from the station's lamps contrasts with the cold, wet environment, highlighting the emotion of the moment.*

Figure S.15: Cinematic styles (Click here to see the video.)

*Graphic Novel type Videos*

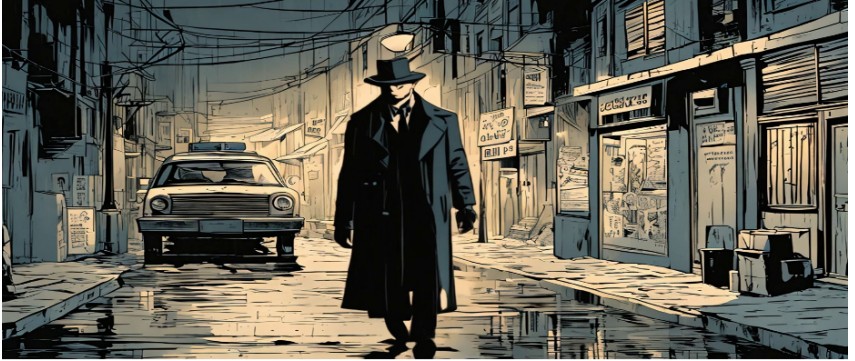

*Create a graphic novel-style video set in a dark, rain-soaked city at night. The camera follows a trench-coated detective as he walks down dimly lit alleyways. The scene is full of sharp contrasts, with deep shadows and bold highlights, capturing the gritty, noir atmosphere. Text bubbles appear with the brief.*

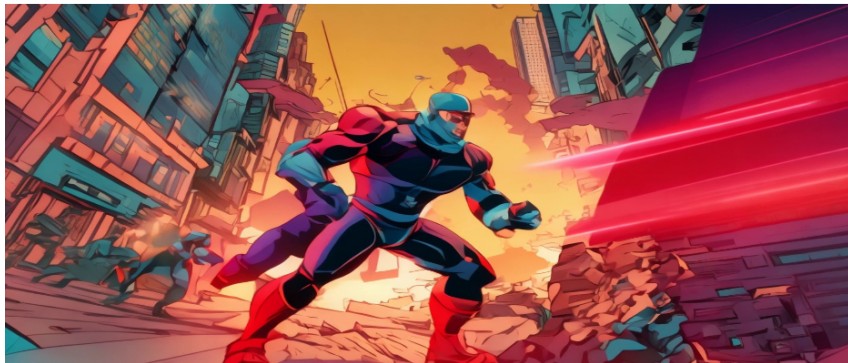

*A graphic novel-style video of a dramatic battle between two superheroes in a futuristic city. The camera zooms in on intense action sequences, with exaggerated motion lines and dynamic angles. Vibrant colors and sharp outlines define the characters, while sound effects like "BAM!" and "CRASH!" are illustrated on screen.*

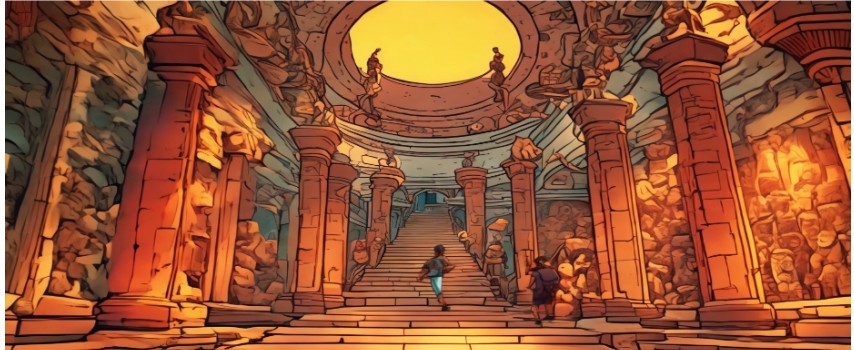

*Create a graphic novel-style video featuring a group of adventurers exploring an ancient, magical temple. The camera captures their journey through booby-trapped corridors and mystical chambers. The scene is filled with intricate, hand-drawn details, vibrant colors, and dramatic lighting.*

Figure S.16: Graphic novel styles (Click here to see the video.)

**Low Polly Videos**

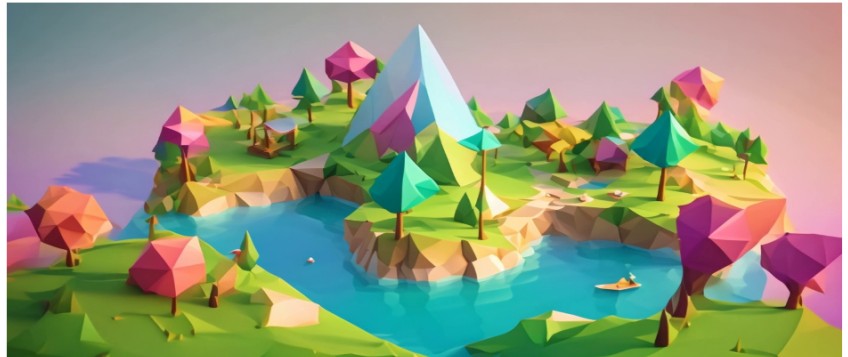

*Create a low poly 3D video featuring a small, vibrant island surrounded by crystal-clear water. The camera flies over the island, revealing its simple, geometric trees, mountains, and a small village. The scene has a playful, colorful aesthetic with sharp, clean edges and minimal detail, evoking a whimsical and adventurous atmosphere.*

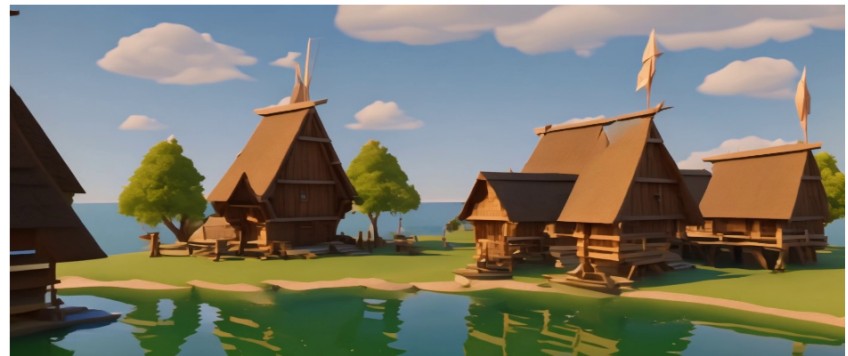

Model a low-poly 3D Viking village, featuring small wooden houses with thatched roofs, a central longhouse, and a wooden dock by the water. Include low-poly boats floating nearby. Keep the terrain flat and the structures simple, using basic polygons.

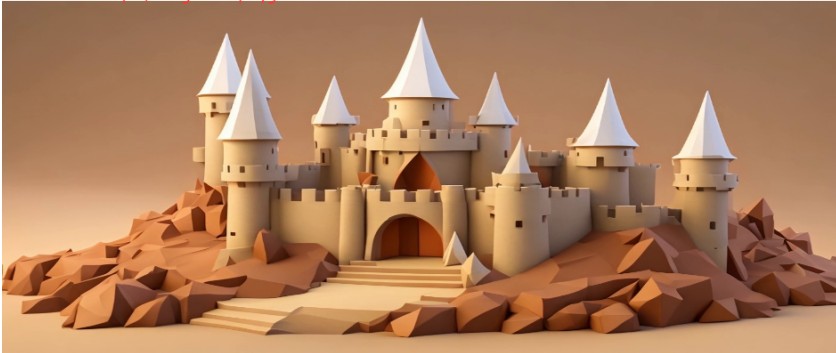

Create a low-poly 3D fantasy castle, featuring towers with cone-shaped roofs, and thick stone walls. Keep the design simple, focusing on basic geometric shapes and solid colors.

Figure S.17: Low Polly results (Click here to see the video.)

*Sci-Fi Videos*

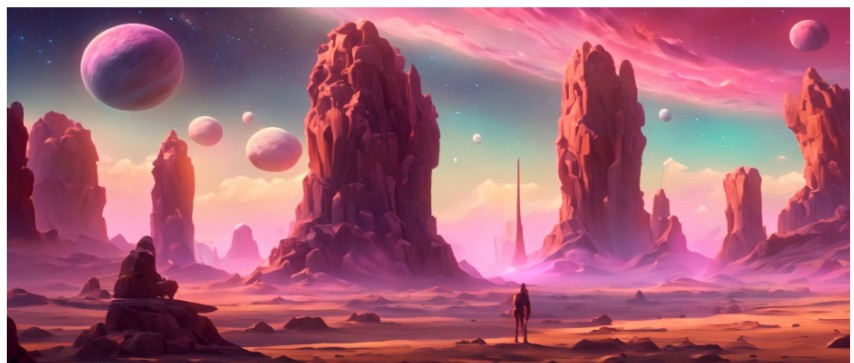

*Create a sci-fi art video featuring a vast, alien landscape on a distant planet. The camera slowly pans across towering, otherworldly rock formations, glowing alien flora, and a sky filled with multiple moons and distant stars. The scene is bathed in surreal colors, with a mix of natural.*

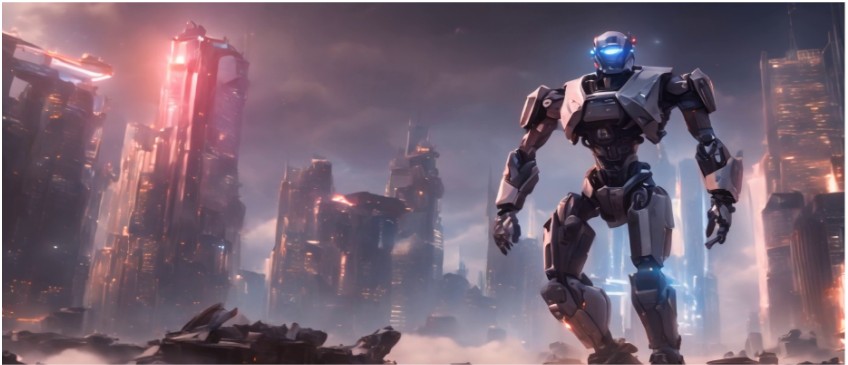

Robots have rebelled against their human creators in a futuristic metropolis. Incorporate visuals of towering skyscrapers.

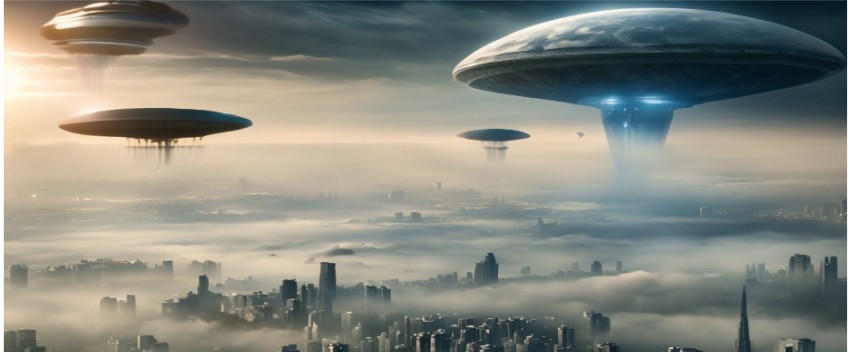

An impending alien invasion. Massive alien motherships loom over major cities, deploying drones and extraterrestrial soldiers.

Figure S.18: Sci-fi results (Click here to see the video.)

**Thriller Videos**

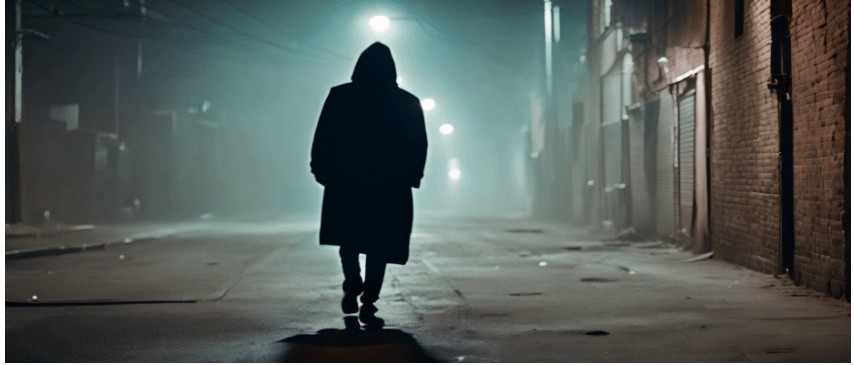

*Thriller-style video set in a dimly lit urban alley at night. The camera follows a lone figure walking quickly, constantly looking over their shoulder. The tension builds with each step as the atmosphere becomes increasingly claustrophobic.*

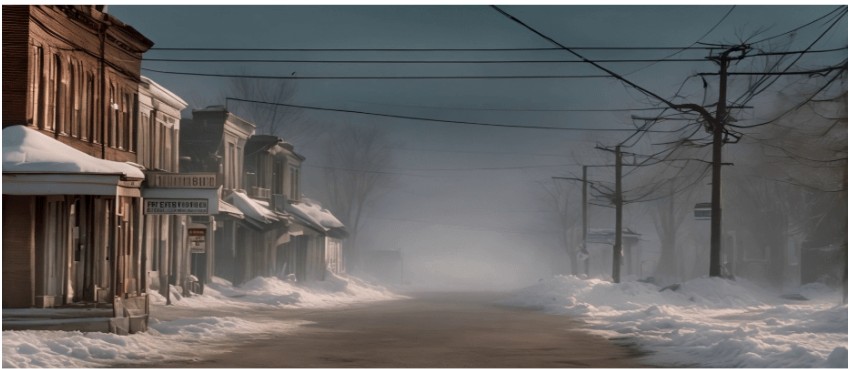

*A small town that starts receiving an unexplained broadcast signal. Incorporate visual glitches and eerie sound effects to enhance the unsettling atmosphere.*

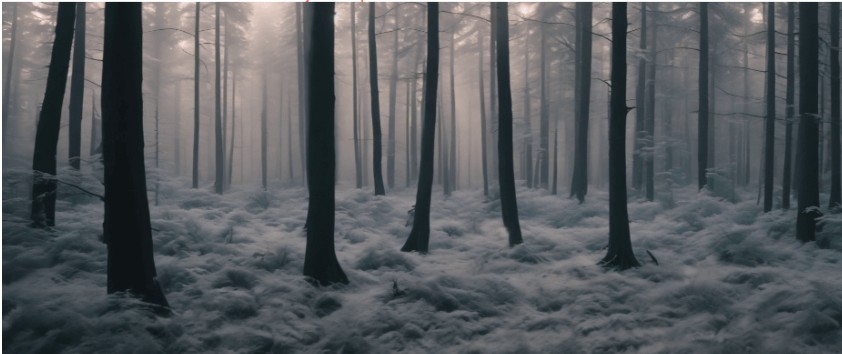

*A dense forest where reality seems distorted, Use atmospheric visuals and disorienting sound design to create a sense of fear and confusion as the environment manipulates their senses.*

Figure S.19: Thriller results (Click here to see the video.)

**Macro Style**

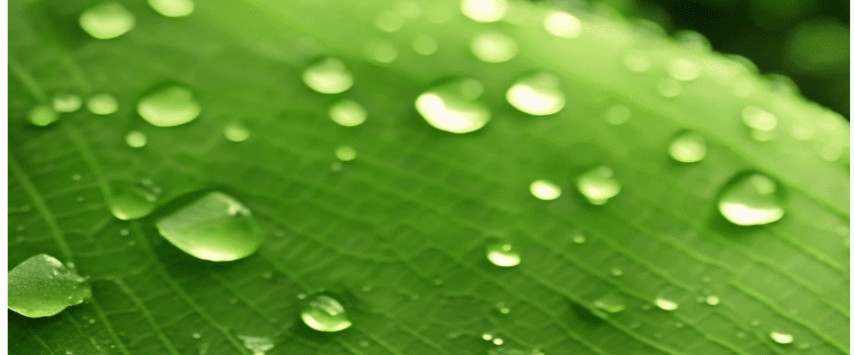

*Render a macro-style video focusing on morning dew droplets on a vibrant green leaf. The camera moves slowly, capturing the crystal-clear droplets as they reflect the surrounding environment. The intricate patterns of the leaf's veins are highlighted, with the sunlight causing the droplets to sparkle like tiny jewels.*

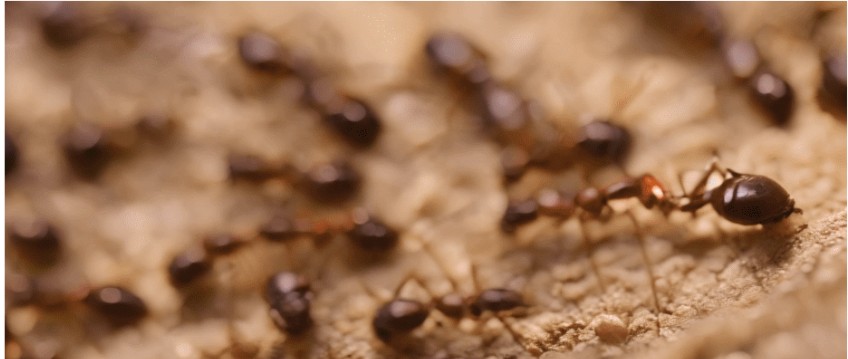

*Create a macro-style video that zooms in on the intricate details of an ant colony at work. The camera captures extreme close-ups of the ants as they carry food and interact with their environment. The textures of their exoskeletons, the fine hairs on their bodies, and the grains of sand beneath them are all vividly detailed.*

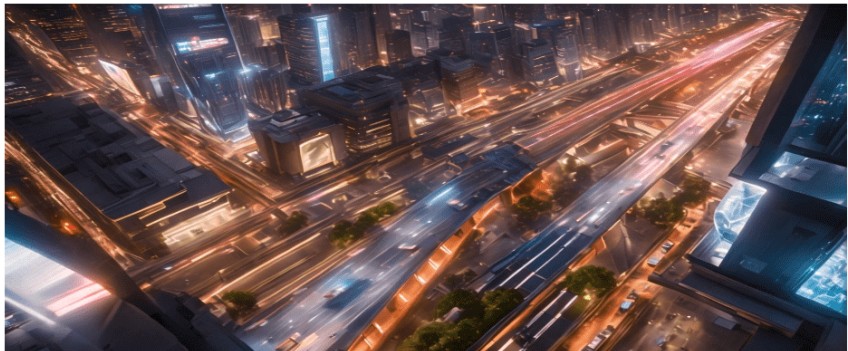

*Produce a fast-paced montage of a futuristic metropolis. Incorporate holographic advertisements, flying vehicles, and interconnected skybridges. The video should feature sharp contrasts between shadows and vibrant lights, with seamless transitions that reflect the sleekness of advanced technology.*

Figure S.20: Macro style results (Click here to see the video.)

