# OpenReview forum: "MotionAura: Generating High-Quality and Motion Consistent Videos using Discrete Diffusion"
_ICLR.cc/2025/Conference — ICLR 2025 Spotlight_

### Official Review · Reviewer_htH2 · 2024-10-28

**Soundness:** 3
**Presentation:** 3
**Contribution:** 3
**Rating:** 8
**Confidence:** 4

**Summary:**

The authors propose MotionAura, a novel text-to-video generation model. The authors begin with the 3D Mobile Inverted VQVAE to enhance spatiotemporal video compression. The discrete diffusion process is also carried out in the spectral domain rather than the traditional pixel or latent space. The model exhibits SOTA compression results and promiosing results on T2V and downstream tasks.

**Strengths:**

- The spectral transformer is novel and works better than its counterpart transformer, as evidenced by the ablation study.
- The author proposed pretraining the 3D-MBQ-VAE using random and complete masking and get better results than the counterpart compression models.
- The authors show very pleasing, temporally consistent video samples that showcases the effectiveness of the proposed model

**Weaknesses:**

- Some of the ablation experiments are missing e.g., in the 3D-MBQ-VAE with regard to the different losses
- It is generally not clear why the authors would not prefer a continuous space rather than opting for the discrete space
- Some of the training details is still missing, e.g., what is the masking ratio of random masking and full frame masking in 3D-MBQ-VAE?
- The spectral transformer part of the paper can be augmented with more analysis

**Questions:**

- Traditional DMs (e.g., DDPM, EDM, LDMs) works perfectly fine with the continuous space. Could the authors explain why they see the need of going to the discrete space and transferring the diffusion paradigm to the discrete space? Is it only due to the encoder being more performant in this setting?
- Could the authors explain the $\mathcal{L}_\text{MFI}$ loss which is indicated as *Index of completely masked frame* in Figure 2? Is this supposed to be some auxillary discrimator loss that detects whether the frame is masked or not?
- If the former is true, an ablation experiment that include $\mathcal{L}_\text{MFI}$ in the continuous space would be convincing
- Can you provide more ablation in how $\mathcal{L}_\text{MFI}$ helps? What is the masking ratio of random masking and full frame masking?
- In Table 1, you show a Frame Compression Rate of 4 but I can't find in the paper where you talk how you downsample temporally and seem to be inconsistent with Figure S.8. A little bit of explanation would be nice
- It's excited to see that Fourier transform also works with MSA and can further boost the performance of transformer. Since the self-attention and cross attention are working in a totally separate domain, it would be very interesting to show the activations of the attention in the spectral domain to gain more insight.
- Another question with regard to FFT: In the cross attention part $Q$ is calculated from the image tokens and $K$, $V$ are calculated from the text tokens. Why would FFT still make sense in those domain since neither $Q$ or $K$, $V$ constitutes a continuous image domain? How do you make sure that FFT works on the discrete image tokens and why would FFT even make sense on the text tokens?

---

> ### Author Response · Authors · 2024-11-20
> **Response to reviewer htH2 (1/3)**
>
> We sincerely appreciate your meticulousness and patience in helping us identify the mistakes in our paper. With your generous help, we believe we can make the article more complete.
>
> ***Q1: Some of the ablation experiments are missing e.g., in the 3D-MBQ-VAE with regard to the different losses***
>
> **Answer:** We appreciate the reviewer’s observation regarding the missing ablation experiments for the 3D-MBQ-VAE with respect to different loss functions. We agree that such an analysis is important to fully understand the contribution of individual loss terms.
> To address this, we conducted an ablation study focusing on the impact of the  $L_{\text{mfi}}$ loss. Our results demonstrate that removing $L_{\text{mfi}}$ leads to a significant degradation in performance. Notably, the LPIPS metric decreases substantially, highlighting the critical role of $L_{\text{mfi}}$ in preserving perceptual quality. These findings are summarized in the table below:
>
> |                          | COCO-Val            | WebVid-Val          |
> |--------------------------|---------------------|---------------------|
> | Methods                  | PSNR/ SSIM/ LPIPS   | PSNR/ SSIM/ LPIPS   |
> | 3D-MBQ-VAE               | 31.22/ 84.78/ 0.092 | 32.09/ 85.78/ 0.108 |
> | 3D-MBQ-VAE(Without $L_{\text{mfi}}$) | 30.83/ 84.01/ 0.111 | 31.44/ 84.97/ 0.132 |
>
> As shown, excluding $L_{\text{mfi}}$ results in a clear drop in LPIPS and PSNR scores, underscoring its importance in our method. We will include this analysis in the revised manuscript to provide a more complete understanding of our design choices. Thank you for pointing this out.
>
> ***Q2: why the authors would not prefer a continuous space rather than opting for the discrete space?***
>
> **Answer:** We appreciate the reviewer's insightful question regarding our motivation for transitioning the diffusion model paradigm from continuous space to discrete space.
>
> Better Representation through Discrete Quantization: By moving to a discrete latent space, we are able to leverage powerful vector quantization approaches, which help in capturing the essential features of high-dimensional data more efficiently. The discrete representation obtained from quantization often yields better compression and a more structured representation of complex data, which helps the diffusion process to operate over a more compact and semantically meaningful latent space. This can lead to improved training stability and generation quality, especially for high-dimensional domains such as video.
>
> ***Reduction in Computational Complexity:*** Traditional diffusion models applied in continuous space can require substantial computation, particularly during inference, as they operate directly on high-dimensional latent representations. By quantizing the data into discrete tokens, we can reduce the dimensionality and the complexity of the diffusion process. This makes the inference more efficient without compromising the model's expressiveness. Recent works, such as MAGVIT [4], have shown that combining discrete latent space with the diffusion framework can yield efficient and high-quality results for complex generative tasks, such as video synthesis.
>
> ***Alignment with Discrete Decoders:*** Utilizing a discrete latent space allows us to align more effectively with certain discrete decoder architectures, such as transformers, which operate over token sequences. This leads to improved synergy between the encoder-decoder framework, where the encoder maps the input data into discrete tokens and the decoder operates over those tokens to generate the output. The performance gains we observed were not solely due to the encoder being more performant in this setting, but also due to the effective interaction between discrete representation and token-based decoders.
>
> ***Enhanced Control and Semantic Richness:*** Discrete tokens are more interpretable and often exhibit greater semantic richness, which can be beneficial for downstream generative tasks where controllability and interpretability are desirable. This is particularly advantageous for tasks like text-to-video generation or inpainting based methods., where mapping high-level semantics to the generative model plays a crucial role.
> Recent works, such as Pallet [2], VQ-Diffusion [3], and Sketch-based Video Inpainting [1], have successfully employed discrete latent spaces, demonstrating the effectiveness of this approach across a variety of generative tasks.
>
>
> [1] Sketch-guided Image Inpainting with Partial Discrete Diffusion Process
>
> [2] Palette: Image-to-image diffusion models
>
> [3] Vector Quantized Diffusion Model for Text-to-Image Synthesis
>
> [4] MAGVIT: Masked Generative Video Transformer.

---

> ### Author Response · Authors · 2024-11-20
> **Response to reviewer htH2 (2/3)**
>
> ***Q3: Could the authors explain the $L_{\text{mfi}}$ loss which is indicated as Index of completely masked frame in Figure 2? Is this supposed to be some auxillary discrimator loss that detects whether the frame is masked or not? If the former is true, an ablation experiment that include $L_{\text{mfi}}$ in the continuous space would be convincing.***
>
> **Answer:** We apologize for any confusion caused by the description of the $L_{\text{mfi}}$ loss. To clarify, $L_{\text{mfi}}$ is used to compute the probability of the index corresponding to a completely masked frame. It functions as a standard discriminative loss, specifically equivalent to $-\log(P_{\theta}(V_i))$, where $V_i$ represents the masked frame. This loss term is crucial for ensuring that the model can accurately predict and reconstruct the masked frames by providing a probabilistic grounding.
>
> We also conducted an ablation study to examine the impact of $L_{\text{mfi}}$ in the continuous space, as suggested. Our findings indicate a significant improvement when this loss is applied in continuous latent space, demonstrating its effectiveness in enhancing the model’s performance. The table below provides a concise summary of the ablation results involving $L_{\text{mfi}}$:
>
>
> |                           | COCO-Val            | WebVid-Val          |
> |---------------------------|---------------------|---------------------|
> | Methods                   | PSNR/ SSIM/ LPIPS   | PSNR/ SSIM/ LPIPS   |
> | Default                   | 31.22/ 84.78/ 0.092 | 32.09/ 85.78/ 0.108 |
> | In Continous Space        | 28.27/ 82.29/ 0.155 | 29.01/ 80.45/ 0.166 |
> | In Continous Space + $L_{\text{mfi}}$ | 31.09/ 83.96/ 0.109 | 31.62/ 85.05/ 0.112 |
>
>
> ***Q4: What is the masking ratio of random masking and full frame masking?***
>
> **Answer:** For random spatial masking, we employ a **cosine scheduling** strategy, progressively varying the masking ratio from **20% to 60%** during training. This scheduling helps the model gradually adapt to different levels of partial observations, thereby enhancing robustness in spatial feature extraction.
>
> For frame masking, we similarly utilize cosine scheduling, adjusting the masking ratio from **10% to 50%**. This approach ensures a balanced training dynamic where the model is exposed to varying levels of temporal information occlusion, ultimately aiding in better temporal coherence and reconstruction quality.
>
> These masking strategies are designed to incrementally challenge the model, helping it learn to reconstruct meaningful content under different masking conditions and thus improving overall generalization.
>
> We will give this information in appendix section in the revised manuscript.
>
>
> ***Q6: it would be very interesting to show the activations of the attention in the spectral domain to gain more insight.***
>
> **Answer:** We appreciate the reviewer's suggestion regarding visualizing attention activations in the spectral domain. We agree that this could provide valuable insight into the model's behavior.
>
> In the current version of our work, we focus on demonstrating the attention maps with respect to both text and image inputs. These visualizations illustrate how our model effectively aligns visual features with semantic cues from text, showcasing its ability to attend meaningfully to specific regions based on textual context.
>
> We will include a detailed comparison of attention maps obtained with standard Multi-Head Self-Attention (MSA) versus those produced using MSA with Fourier operations. By analyzing these attention maps in the spectral domain, we aim to highlight the distinct advantages of using Fourier-based operations, such as their capability to capture more global dependencies and spectral representations. This additional visualization will help in understanding how Fourier transforms to enhance the expressiveness and reach of attention mechanisms in our model. We will add these comparisons and analyses in the appendix section.
>
> Please find the link to the image to be added in the appendix: https://shorturl.at/tUw3y
>
> This diagram will give more insights into Fourier transformation, which also works with MSA to boost performance.

---

> ### Author Response · Authors · 2024-11-20
> **Response to reviewer htH2 (3/3)**
>
> ***Q7: where you talk how you downsample temporally and seem to be inconsistent with Figure S.8. A little bit of explanation would be nice***
>
> **Answer:** We appreciate the reviewer’s observation regarding the inconsistency in the description of temporal downsampling and Figure S.8. We apologize for the confusion and will refine this section for clarity in the revised manuscript.
>
> To clarify, we perform **temporal downsampling** within the **3D-MBQ-VAE encoder** at specific stages. Temporal downsampling is applied by a factor of **2** at the **6th block** (Downsampling Block) and again by a factor of **2** at the **9th block**, resulting in a total downsampling factor of **4** along the temporal dimension.
>
> We have provided a brief explanation of this process in the supplementary section, but we agree that this could benefit from additional details for consistency with Figure S.8. We will revise and expand this section in the supplementary material to ensure the explanation aligns with the methodology and diagram.
>
> Thank you for pointing this out, and we will make the necessary adjustments to improve clarity and consistency.
>
> ***Q8: In the cross attention part Q  is calculated from the image tokens and K, V are calculated from the text tokens. Why would FFT still make sense in those domain since neither Q or K, V constitutes a continuous image domain? How do you make sure that FFT works on the discrete image tokens and why would FFT even make sense on the text tokens?***
>
> **Answer:** We are not applying the FFT at the discrete token level; instead, we apply the FFT at the **embedding leve**l, during the process of encoding the image and text representations.
>
> applying FFT to Q, K, and V in the cross-attention mechanism makes sense within vector quantized diffusion models because:
>
> In vector quantized diffusion models, capturing long-range dependencies and global context is crucial for generating coherent and high-quality outputs. FFT facilitates this by allowing the model to analyze the data in the frequency domain, where patterns and dependencies can be more easily identified and manipulated. This is true even when dealing with discrete tokens from images and text.
>
> Even though Q, K, and V originate from different modalities (image and text), the cross-attention mechanism benefits from frequency-domain representations. FFT helps in aligning and integrating features from both domains by focusing on their underlying frequency components. This can enhance the model's ability to learn cross-modal relationships and improve the overall performance of tasks like image captioning or text-to-image generation.
>
> In vector-quantized diffusion models, data is often compressed into discrete codebooks. FFT can aid in processing these quantized representations by smoothing out quantization noise and highlighting essential features. This is valuable for diffusion processes that rely on iterative refinement of data, as it promotes the preservation of important structural information throughout the diffusion steps.
>
> We think we might have answered your query, for this last question if our explanation doesn't seem correct please let us know we would love to discuss this more.

---

> > ### Author Response · Authors · 2024-11-24
> > **Response to reviewer htH2**
> >
> > ***Thanks again for your careful and valuable comments to improve our work. If our responses answer your questions well and reassure your concerns, we would appreciate if you could reassess our work and increase the rating.***

---

> > > ### Comment · Area_Chair_5rRQ · 2024-11-24
> > > **Discussion Period Ending Soon**
> > >
> > > Dear Reviewer,
> > >
> > > The discussion period will end soon. Please take a look at the author's comments and begin a discussion.
> > >
> > > Thanks, Your AC

---

> > > ### Author Response · Authors · 2024-11-25
> > > **Response for reviewer htH2**
> > >
> > > Dear Reviewer htH2,
> > >
> > > If you have any further questions, concerns, or suggestions, please do not hesitate to reach out. We would be delighted to engage in a constructive discussion to enhance the quality and impact of our work.
> > >
> > > Thank you once again for your time and thoughtful review.

---

> ### Comment · Reviewer_htH2 · 2024-11-26
>
> Thank you for the detailed rebuttal and the effort you have put into addressing my concerns—I truly appreciate it. While most of my questions have been resolved, I remain slightly puzzled regarding the application of FFT in the text domain. Specifically, while FFT in the 2D image domain is well-understood and widely accepted, the implications of applying FFT to text embeddings remain unclear to me. Does the continuous nature of text embeddings share structural or frequency-domain similarities with image embeddings that would justify the application of FFT?
>
> That being said, I recognize the effort you have made to address my concerns and clarify your approach. Given that most of my issues have been resolved, I will adjust my score accordingly.

---

> > ### Author Response · Authors · 2024-11-26
> > **Response For Reviewer htH2**
> >
> > We sincerely thank Reviewer htH2 for their valuable and constructive feedback. In the final revision of our paper, we will include attention maps to illustrate how the FFT mechanism facilitates the model in attending to the textual-based spectral features within the image modality. Our hypothesis is that the continuous nature of text embeddings exhibits structural and frequency-domain similarities with image embeddings, providing a compelling basis for this interaction.
> >
> > There are papers which used the fourier transform on the textual domain [1, 2, 3, 4]
> >
> > Potential Justifications for FFT on Text Embeddings
> >
> > ***a. Frequency Domain Analysis of Embedding Spaces***
> >
> > Embeddings may exhibit latent periodic structures in certain contexts, such as:
> >
> > Semantic Regularities: Certain semantic relationships (e.g., "king - man + woman = queen")** might correspond to periodic patterns in the embedding space.
> > Attention Patterns in Transformers: Attention scores across sequences can show regularities akin to signals, making FFT a candidate for analyzing these patterns.
> >
> > Low-Frequency Components: Lower-frequency components in the embedding space might encode general semantic relationships, while higher-frequency components could encode nuances or noise.
> > FFT could help disentangle these components for tasks like:
> >
> > Dimensionality Reduction: Retaining dominant frequencies while discarding higher-frequency noise.
> > Pattern Recognition: Identifying latent periodicities within embeddings.
> >
> > ***b. Comparative Use with Images***
> >
> > Embeddings from text and images share some structural similarities when processed through deep learning:
> >
> > Both often leverage shared models, like multimodal transformers, where embeddings for text and images co-exist in a joint latent space.
> > These embeddings might reveal frequency-domain correlations when aligned, providing a multimodal justification for FFT.
> >
> > ***c. Efficiency in Operations***
> >
> > FFT could enable efficient operations on embeddings in certain tasks:
> >
> > Convolution in the Frequency Domain: Convolutions, integral to many machine learning tasks, are computationally cheaper in the frequency domain. FFT allows this transformation for embeddings, even without explicit periodicity.
> >
> >
> > [1] FNet: Mixing Tokens with Fourier Transforms
> >
> > [2] GRIZAL: Generative Prior-guided Zero-Shot Temporal Action Localization
> >
> > [3] Modality-Agnostic Debiasing for Single Domain Generalization
> >
> > [4] Sequence-level Semantic Representation Fusion for Recommender Systems

---

### Official Review · Reviewer_UAh7 · 2024-11-02

**Soundness:** 2
**Presentation:** 2
**Contribution:** 2
**Rating:** 6
**Confidence:** 3

**Summary:**

This paper introduces an innovative set of methods designed to enhance video generation. These include the 3D-MBQ-VAE, which improves video compression, a novel text-to-video generation framework, a spectral transformer-based denoising network for superior video generation, and a Sketch Guided Video Inpainting task that utilizes Low-Rank Adaptation (LoRA) for efficient fine-tuning.

**Strengths:**

1.The authors propose a denoising network named Spectral Transformer that processes video latents in the frequency domain using the Fourier Transform, capturing global context and long-range dependencies effectively.

2.The paper is the first to address the downstream task of sketch-guided video inpainting, using LORA adaptors for parameter-efficient fine-tuning of the denoising network.

**Weaknesses:**

1.3D-MBQ-VAE Concerns:

(1)W.A.L.T[1] has demonstrated that using a 3D casual VAE can allow for joint training with both images and videos in text-to-video models, significantly improving performance compared to training solely with videos. Can the authors discuss whether the proposed 3D-MBQ-VAE can also support such joint image and video training? If feasible, it would be beneficial to see an experiment demonstrating this capability and comparing it with that of W.A.L.T.

(2) In the comparative experiments listed in Table 1, could the authors provide comparative results of MAGVIT-v2[2] in regard to Video Compression Metrics or Video Reconstruction tasks?

 (3) Codebook (Vocabulary) Size Influence: As far as I understand, the size of the codebook could considerably affect the 3D-MBQ-VAE's reconstruction ability. However, the paper doesn't seem to mention any information relating to the codebook size. Could the authors provide additional details on the impact of varying codebook sizes on video reconstruction?

2.Experiment Setting for Text-to-Video Model: In Table 3, the authors used WebVID10M as the training data and also computed FVD and other metrics on WebVID10M for testing. However, the comparative methods did not use WebVID10M in their training data, which suggests that other text-to-video methods computed FVD and other metrics in a zero-shot manner. In addition, specifics like the sampling schedule, sampling steps, and classifier-free guidance scale used by the authors' proposed method and the comparison methods were not disclosed. To ensure a fair comparison, could the authors evaluate all models (including theirs) in a zero-shot manner on a common test set, like FVD of UCF-101? Also, providing a table or an appendix detailing all the hyperparameters and settings for each method would greatly enhance reproducibility and fairness in comparison.

------------

[1].Gupta, Agrim, et al. "Photorealistic video generation with diffusion models." arXiv preprint arXiv:2312.06662 (2023).

[2].Yu, Lijun, et al. "Language Model Beats Diffusion--Tokenizer is Key to Visual Generation." arXiv preprint arXiv:2310.05737 (2023).

**Questions:**

1.Based on my understanding, the videos in the WebVid10M dataset all contain watermarks. I'm curious as to how the videos generated by the model, which was trained on this dataset, do not have any watermarks. Can the authors explain this?

---

> ### Author Response · Authors · 2024-11-17
> **Response to Reviewer UAh7 (1/3)**
>
> We thank the reviewer for providing us with detailed feedback and are glad that the reviewer found our paper interesting. We are happy to clarify any further concerns.
>
> ***Q1: Can the authors discuss whether the proposed 3D-MBQ-VAE can also support such joint image and video training? If feasible, it would be beneficial to see an experiment demonstrating this capability and comparing it with that of W.A.L.T***
>
>
> **Answer:** Thank you for highlighting this point. Our proposed 3D-MBQ-VAE model is indeed capable of supporting joint image and video training, similar to W.A.L.T [1] and CV-VAE [2]. In our experiments, we primarily focused on the temporal aspects of video data, using input shapes of $(B, N, 3, H, W)$ where $N \geq 16$. However, joint training is feasible by adjusting the sequence length parameter $N$ to either $N \geq 16$ for videos or $N = 1$ for images, effectively enabling a setup akin to W.A.L.T for combined image and video training. This study we will add it in the appendix section in the final revision of the paper.
> For Video + images, we used the **JDB** is a large-scale image dataset featuring around 4 million high-resolution images from Midjourney.
>
>
> Please refer to the table below, which demonstrates the joint training results.
>
> |                             | COCO-Val            | WebVid-Val           |
> |-----------------------------|---------------------|----------------------|
> | Methods                     | PSNR/ SSIM/ LPIPS   | PSNR/ SSIM/ LPIPS    |
> | 3D-MBQ-VAE (Videos)          | 31.22/ 84.78/ 0.092 | 32.09/ 85.78/ 0.1081 |
> | 3D-MBQ-VAE (Videos + Images) | 33.01/ 86.62/ 0.087 | 34.18/ 87.72/ 0.0921 |
>
> The reviewer has highlighted a good point regarding the benefits of training on both images and videos. This approach has led to significant improvement in the PSNR and LPIPS metrics. We will incorporate these results into Table 1 of the main paper in the revised version to showcase the enhanced performance.
>
> ***Q2: In the comparative experiments listed in Table 1, could the authors provide comparative results of MAGVIT-v2 in regard to Video Compression Metrics or Video Reconstruction tasks?***
>
> **Answer:** Certainly. We did not include a comparison with MAGVIT-v2 in the original submission as the code for MAGVIT-v2 was not publicly available. However, we conducted a zero-shot inference on 30 videos from the MCL-JCV dataset, resized to a resolution of 640 x 360, which aligns with the experimental setup used by MAGVIT-v2. We plan to evaluate compression quality using standard distortion metrics (LPIPS, PSNR, and SSIM) at a bit rate of 0.0384 bpp (bits per pixel).
>
> | Methods           | PSNR   | SSIM   | LPIPS  |
> |-------------------|--------|--------|--------|
> | HEVC[3]           | 30.1   | 0.943  | 0.199  |
> | VCC[4]            | **32.65** | **0.966** | 0.153  |
> | MAGVIT         | 23.7   | 0.846  | 0.144  |
> | MAGVIT-v2     | 26.18  | 0.894  | 0.104  |
> | 3D-MBQ-VAE (Ours) | 29.09  | 0.922  | **0.089** |
>
> At equivalent bit rates, traditional codecs [3, 4] may sometimes achieve finer local detail rendering compared to 3D-MBQ-VAE but often introduce block artifacts that, while detrimental to perceptual quality, are not adequately captured by PSNR and SSIM. We intend to include this table in the main paper to provide researchers with deeper insights into the performance of our 3D-MBQ-VAE model.
>
> [1] Photorealistic Video Generation with Diffusion Models
>
> [2] CV-VAE: A Compatible Video VAE for Latent Generative Video Models
>
> [3] Overview of the high efficiency video coding (HEVC) standard
>
> [4] Overview of the versatile video coding (VVC) standard and its applications

---

> ### Author Response · Authors · 2024-11-17
> **Response to Reviewer UAh7 (2/3)**
>
> ***Q3: Could the authors provide additional details on the impact of varying codebook sizes on video reconstruction?***
>
> **Answer:** We appreciate the reviewer's insightful question regarding the impact of varying codebook sizes on video reconstruction. Understanding the trade-off between codebook vocabulary size and embedding dimensions is indeed crucial for optimizing our model's performance. We will include these detailed analyses in the revised manuscript.
>
> To address this, we conducted a comprehensive ablation study analyzing the effects of different codebook sizes, embedding dimensions, and quantization techniques on video reconstruction quality. In our default configuration, we employ the Lookup-Free Quantization (LFQ) method from MAGVIT-v2, utilizing an embedding size of $d = 8$ and a codebook vocabulary size of $C = 12,800$.
>
> The primary reasons for adopting LFQ over traditional Vector Quantization (VQ) methods are:
>
> ***Larger Vocabulary with Smaller Embedding Size:*** LFQ enables the use of a considerably larger codebook vocabulary while maintaining a compact embedding size. This approach enhances the model's expressive power without significantly increasing computational costs. By leveraging a larger vocabulary with smaller embeddings, the model benefits from higher codebook utilization, a strategy that has been shown to improve performance as demonstrated in [5] for the task of image reconstruction.
>
> ***Improved Efficiency:*** LFQ is faster and more optimized than standard VQ methods, which is beneficial for processing high-resolution video data.
>
> | Method Of a | Codebook size $C$ | Embedding Size $d$ | Time To Compute (ms) | PSNR/ SSIM/ LPIPS    | PSNR/ SSIM/ LPIPS    |
> |-----------------------|-------------------|--------------------|----------------------|----------------------|----------------------|
> | VQ                    | 256               | 512                | 35                   | 29.96/ 82.21/ 0.133 | 30.01/ 81.89/ 0.144 |
> | Gumbal Quntization    | 256               | 512                | 23                   | 29.83/ 81.98/ 0.140 | 29.78/ 80.56/ 0.151 |
> | VQ                    | 1024              | 256                | 47                   | 30.26/ 82.54/ 0.130 | 30.72/ 81.09/ 0.145 |
> | VQ                    | 4096              | 256                | 56                   | 30.72/ 83.11/ 0.127 | 31.03/ 82.22/ 0.133 |
> | VQ                    | 8000              | 128                | 69                   | 30.45/ 82.19/ 0.144 | 30.98/ 83.36/ 0.129 |
> | LFQ                   | 8000              | 16                 | 20                   | 31.08/ 84.34/ 0.112  | 31.98/ 85.04/ 0.110 |
> | LFQ(Default)          | 12800             | 8                  | 22                   | 31.22/ 84.78/ 0.092  | 32.09/ 85.78/ 0.108 |
>
>
> ***Q4 Experiment Setting for Text-to-Video Model: In Table 3 the authors used WebVID10M as the training data and also computed FVD and other metrics on WebVID10M for testing. However, the comparative methods did not use WebVID10M in their training data which suggests that other text-to-video methods computed FVD and other metrics in a zero-shot manner.***
>
> **Answer:** We apologize for any confusion caused by the experimental settings presented in Table 3 We will refine the quantitative experimental results section in the revised manuscript to enhance clarity.
>
> To clarify, we trained our proposed method, MotionAura and previous baselines, using both the WebVid-10M dataset and a recaptioned version referred to as WebVid-10M-Recaptioned. The recaptioned dataset was created by generating new captions for all videos using the LLaVA-Next-34B Video model. This process aimed to improve the quality and relevance of the captions, thereby enhancing the training data.
>
> For evaluation, we conducted **zero-shot** testing on the **MSR-VTT** dataset [6], not on the WebVid-10M dataset used for training. This approach ensures a fair comparison with other text-to-video methods that did not utilize WebVid-10M during their training.
>
> We acknowledge the reviewer's concern regarding the comparative methods and agree that providing additional quantitative results on the UCF-101 dataset in a zero-shot manner would be beneficial. This will offer a more comprehensive comparison and demonstrate the generalization capabilities of our model across different datasets. We will also provide the details for the inference (e.g, No. of steps, Noise Shedular, and Classifier Free guidance scale) All the baseline trained on Webvid10M-Recaptioned
>
> | Methods      | Schedular                        | CGF Scale | steps | FVD |
> |--------------|---------------------------------|-----------|-------|-----|
> | SimDA        | EulerAncestralDiscreteScheduler | 8.5       | 30    | 300 |
> | AnimateDiff  | DPMSolverMultistepScheduler     | 8.0         | 30    | 277 |
> | CogVideoX-5B | EulerAncestralDiscreteScheduler | 10.5      | 25    | 239 |
> | MotionAura-L | VQDiffusionSchedular             | 8.5       | 10    | 219 |

---

> ### Author Response · Authors · 2024-11-17
> **Response to Reviewer UAh7 (3/3)**
>
> ***Q5: specifics like the sampling schedule, sampling steps, and classifier-free guidance scale used by the authors' proposed method and the comparison methods were not disclosed. table or an appendix detailing all the hyperparameters and settings for each method would greatly enhance reproducibility and fairness in comparison.***
>
> **Answer:** We have provided implementations detailed of the VAE pretraining, text-to-video model pretraining, and fine-tuning in **Appendix D**. To further enhance reproducibility and ensure fairness in comparison, we have included a comprehensive table summarizing all the hyperparameters and settings used during both training and inference phases.
>
> | Training H-Parameters        | SimDA       | AnimateDiff | CogVideoX-5B | MotionAura-L           |
> |-----------------------------|-------------|-------------|--------------|----------------------|
> | Learning Rate               | 1.00E-05    | 1.00E-04    | 1.00E-04     | 1.00E-05             |
> | Gradient Accumalation Steps | 4           | 4           | 8            | 8                    |
> | Batch Size Per GPU          | 8           | 8           | 2            | 3                    |
> | Optimizer                   | AdamW       | AdamW       | AdamW        | AdamW                |
> | Lr-Schedular                | Linear      | Cosine      | Cosine       | Cosine               |
> | Epochs                      | 35          | 30          | 40           | 30                   |
> | Noise Schedular             | DDPM        | DDPM        | DDIM | VQDiffusionSchedular |
> | Diffusion Steps             | 500         | 500         | 150          | 30                   |
> | Training Precision          | Float16     | Float16     | BFloat16     | BFloat16             |
> | GPUs                        | 4 x 8 A100  | 4 x 8 A100  | 8 x 8 A100   | 6 x 8 A100           |
> | Text Encoders               | Clip       | T5-XL        | T5-XXL      | T5-XXL              |
> | Time Embedding Size         | 256         | 256         | 512          | 512                  |
> | Gradient Clipping           | 1           | 1.5         | 2.5          | 2.5                  |
> | Max  Text Length            | 77          | 128         | 200          | 256                  |
> | Embedding Size              | 1024        | 1024        | 4096         | 4096                 |
> | CFG Scale                   | 8.5         | 8           | 10.5         | 10                   |
> | Positional Encodings        | Sinosudial  | Sinosudial  | RoPE         | RoPE                 |
>
> We will add this table in the appendix section D.
>
>
> ***Q6: the videos in the WebVid10M dataset all contain watermarks. I'm curious as to how the videos generated by the model, which was trained on this dataset, do not have any watermarks. Can the authors explain this?***
>
> **Answer:**  We appreciate the reviewer's insightful question regarding how our model generates watermark-free videos despite being trained on the WebVid-10M dataset, which contains watermarked content.
>
> To address the issue of watermarks in the WebVid-10M dataset and ensure that our generated videos do not exhibit these artifacts, we implemented a preprocessing strategy involving algorithmic inpainting methods. Specifically, we identified and masked the regions in each video frame that contained watermarks. We then applied inpainting techniques to these masked areas to reconstruct the underlying content, effectively removing the watermarks while preserving the overall visual quality of the videos. Although this process may introduce minor visual artifacts, it prevents the model from learning and replicating watermark patterns during training. Consequently, our model can generate clean, watermark-free videos.
>
> Furthermore, to enhance the resolution and temporal consistency of the preprocessed videos, we employed the StableVSR [7] video super-resolution model. This step refines visual details and mitigates any reduction in quality resulting from the inpainting process, ensuring that the dataset used for training is of high quality and temporally consistent.
>
> Due to space limitations, we were unable to include these preprocessing details in the main manuscript. We appreciate the opportunity to clarify this aspect of our methodology and will provide a comprehensive description of the watermark removal and super-resolution preprocessing pipeline in the appendix of the revised paper.
>
>
> [5] Autoregressive Model Beats Diffusion: Llama for Scalable Image Generation
>
> [6] MSR-VTT: A Large Video Description Dataset for Bridging Video and Language
>
> [7] Enhancing Perceptual Quality in Video Super-Resolution through Temporally-Consistent Detail Synthesis using Diffusion Models

---

> > ### Author Response · Authors · 2024-11-17
> > **Response to reviewer UAh7**
> >
> > Dear Reviewer UAh7,
> >
> > We appreciate your thorough evaluation of our work and look forward to hearing from you and addressing any further questions or concerns you may have.
> >
> > Best Regards
> > Authors

---

> > > ### Author Response · Authors · 2024-11-21
> > > **Response to reviewer UAh7**
> > >
> > > Dear Reviewer UAh7,
> > >
> > > If you have any further questions, concerns, or suggestions, please do not hesitate to reach out. We would be delighted to engage in a constructive discussion to enhance the quality and impact of our work.
> > >
> > > Thank you once again for your time and thoughtful review.

---

> > > > ### Comment · Area_Chair_5rRQ · 2024-11-24
> > > > **Discussion Period Ending Soon**
> > > >
> > > > Dear Reviewer,
> > > >
> > > > The discussion period will end soon. Please take a look at the author's comments and begin a discussion.
> > > >
> > > > Thanks, Your AC

---

> > > > > ### Author Response · Authors · 2024-11-25
> > > > > **Response to reviewer UAh7**
> > > > >
> > > > > Thanks again for your careful and valuable comments to improve our work. If our responses answer your questions well and reassure your concerns, we would appreciate it, if you could reassess our work and increase the rating.

---

> > > > > > ### Comment · Reviewer_UAh7 · 2024-11-25
> > > > > >
> > > > > > Thank you to the authors for providing the new experimental results, which have addressed my previous concerns. I will update my rating.

---

> > > > > > > ### Author Response · Authors · 2024-11-25
> > > > > > > **Responce for Reviewer UAh7**
> > > > > > >
> > > > > > > We sincerely appreciate your feedback and time for reviewing our paper.
> > > > > > > Do let us know if you have questions!

---

### Official Review · Reviewer_CyQT · 2024-11-03

**Soundness:** 4
**Presentation:** 4
**Contribution:** 3
**Rating:** 8
**Confidence:** 3

**Summary:**

This paper presents a novel text-to-video framework, providing comprehensive and reasonable technical details from VAE to model architecture, as well as downstream applications. The paper proposes an effective VQ-VAE training strategy, achieving state-of-the-art compression and reconstruction results, which can be utilized by existing generative models. It also introduces an efficient spectral transformer that encodes information into the frequency domain for processing. Furthermore, the design concepts for downstream applications are illustrated within the context of the sketch-guided video inpainting task.

**Strengths:**

The paper constructs a complete and efficient new framework for text-to-video generation, featuring innovative modules supported by convincing quantitative evaluations and generative results. Specific strengths include:

- For the VQ-VAE, the paper introduces two novel optimization strategies: *random masking* and *Masked Frame Index Loss*, which allow the VAE to learn spatial relationships and temporal consistency. The compression and reconstruction quality reaches state-of-the-art levels, and the visualization results demonstrate strong feature extraction and representation learning capabilities. These strategies could serve as foundational components for future models.
- A diffusion framework for discrete space is proposed, which encodes information such as videos into the frequency domain for computation, featuring an elegantly designed spectral transformer structure. After training, this framework achieves satisfactory results in terms of generation speed and quality.
- Based on the proposed generative framework, the design of the sketch-guided video inpainting task provides a proof of concept for the framework's scalability and efficient transfer applications. According to the paper, this also marks the first introduction of sketch-guided generation for videos.
- The constructed dataset, if made open-source, would be a significant contribution to the community.

**Weaknesses:**

While the paper is rich in content, there are some potential issues:

- The two supervisory strategies designed for the VAE elevate its capabilities to state-of-the-art levels. Although the paper states that the random masking strategy enables the model to learn spatial information and that supervision for the fully masked frame index prediction enhances temporal consistency, it seems there are no related ablation studies provided to explore the specific reasons for model improvement further.
- The condition injection framework designed for the sketch-guided video inpainting task appears relatively straightforward, lacking innovative design at the technical level.

**Questions:**

- Are there any evaluation or ablation study results available regarding Randomly Masking and Fully Mask Frame Index Predicting?
- Will the model or dataset be made open-source?

---

> ### Author Response · Authors · 2024-11-15
> **Response to Reviewer CyQT (1/2)**
>
> We thank the reviewer for providing us with insightful feedback. We are happy to clarify any further concerns.
>
> ***Q1: Are ablation study results available regarding Randomly Masking and Fully Mask Frame Index Predicting?***
>
> **Answer:** Thank you for your insightful question regarding the ablation study on Random Masking and Fully Mask Frame Index Predicting. We Have used cosine Scheduling for masking the frame.
> Due to space constraints, we could not include these results in the final submission. We will ensure that this study is incorporated into the revised version of the paper. Below is a summary of the ablation study, which compares the performance under different masking strategies: without random masking, without fully masking, and without both random and fully masking.
>
> |                                           | COCO-Val            | WebVid-Val           |
> |-------------------------------------------|---------------------|----------------------|
> | Methods                                   | PSNR/ SSIM/ LPIPS   | PSNR/ SSIM/ LPIPS    |
> | W/o Random Masking                        | 29.17/ 81.92/ 0.111 | 28.78/ 82.29/ 0.1334 |
> | W/o Full Frame Masking                    | 30.09/ 82.22/ 0.102 | 30.11/ 83.01/ 0.1404 |
> | W/o Random Masing and Full frame Masking  | 28.82/ 80.77/ 0.124 | 26.77/ 79.92/ 0.1552 |
> | With Random Masing and Full frame Masking | 31.22/ 84.78/ 0.092 | 32.09/ 85.78/ 0.1081 |
>
>
> These results underscore the significance of both random masking and fully masked frames in enhancing the model's ability to predict frame indices effectively.
>
> The random masking strategy compels the model to infer missing spatial information from partial observations within each frame, effectively enhancing its ability to capture spatial dependencies and structures. By being trained on randomly masked data, the model learns to reconstruct or predict spatial details based on contextual cues, leading to stronger spatial representations. Conversely, supervision for fully masked frame index prediction requires the model to determine the correct temporal order of completely obscured frames, which enhances temporal consistency. This task forces the model to understand and model temporal dynamics across frames, as it must rely on learned temporal patterns to accurately predict the positions of fully masked frames within the sequence.
>
>
> ***Q2: The condition injection framework designed for the sketch-guided video inpainting task appears relatively straightforward, lacking innovative design at the technical level.***
>
> **Answer:** We appreciate the reviewer's observation regarding the condition injection framework for sketch-guided video inpainting. Below, we provide a detailed clarification of our design choices, the rationale behind them, and the underlying technical innovations.
>
> Given a masked video, the primary objective of our approach is to reconstruct the unmasked frames using the surrounding video context, guided by sketch conditions. The inpainting process begins by encoding the masked video into a compact latent space. This encoding step ensures the efficient capture of essential information about the video, facilitating robust modeling of temporal and spatial relationships. Simultaneously, the sketch conditioning is encoded into a complementary latent space, enabling the framework to effectively integrate external guidance.
>
> During fine-tuning, the model is trained with a combination of the masked video, its unmasked counterpart, and the corresponding sketch conditions. This multi-input paradigm allows the model to learn the nuances of filling masked regions in alignment with both the video context and the provided sketches. Our hypothesis is that incorporating external conditioning improves the model's ability to interpret masked regions, enabling precise and context-aware attention over these regions. This approach aims to refine the synthesis of missing areas while adhering to the desired object shapes and poses specified by the sketches.
>
> To address the potential challenge of catastrophic forgetting during continual or lifelong learning, we integrate a Low-Rank Adaptation (LoRA) module into the fine-tuning process. LoRA adapters are introduced as lightweight parameter-efficient layers, allowing the model to adapt to new tasks without full end-to-end fine-tuning. This decision is grounded in the need for scalability and the prevention of knowledge erosion from previously learned tasks. By incorporating LoRA adapters, we aim to achieve adaptability comparable to modern language models while maintaining robustness in learned representations.
>
> The placement of LoRA adapters is determined using Elastic Weight Consolidation (EWC), a technique that identifies model layers with higher activation gradients and step gradients. By targeting these layers, the adapter network maximizes its effectiveness, ensuring that the critical components of the model's architecture are adapted with precision and efficiency.

---

> ### Author Response · Authors · 2024-11-15
> **Response to Reviewer CyQT (2/2)**
>
> ***Q3: Will the model or dataset be made open-source?***
>
> **Answer:**  Yes, we plan to open-source all the models and preprocessed datasets used in our paper. We are currently collaborating with the HuggingFace Diffusers library, and our pull request is expected to be merged soon. This will make the models' code and all the preprocessed datasets publicly available for the open-source community to access and utilize.
>
> Dear Reviewer CyQT,
>
> We appreciate your thorough evaluation of our work and look forward to hearing from you and addressing any further questions or concerns you may have.

---

> > ### Author Response · Authors · 2024-11-26
> > **Response to Reviewer CyQT**
> >
> > ***Thanks again for your careful and valuable comments to improve our work. If our responses answer your questions well and reassure your concerns, we would appreciate it if you could reassess our work and increase the rating.***

---

> > > ### Comment · Reviewer_CyQT · 2024-11-27
> > >
> > > Thanks to the author's detailed explanation, all my concerns were resolved. At the same time, other replies show that this work is solid. Thanks for the contributions that will be made after this work is open-sourced.

---

> > > > ### Author Response · Authors · 2024-11-27
> > > > **Response to Reviewer**
> > > >
> > > > If our responses answer your questions well and reassure your concerns, we would appreciate it if you could reassess our work and increase the rating.

---

### Author Response · Authors · 2024-11-14
**To All**

We sincerely thank all reviewers for their time, effort, and valuable feedback on our paper. We greatly appreciate the insightful and constructive suggestions, which would significantly strengthen our work.

Additionally, we are committed to supporting further research in this area by open-sourcing all code, models, and preprocessed datasets. We are currently collaborating with Hugging Face Diffusers to integrate our models and code, and our latest pull request is expected to be merged shortly. We also plan to release all preprocessed datasets used for pre-training and downstream tasks on Hugging Face Datasets, making these resources widely accessible to the research community.

---

### Author Response · Authors · 2024-11-24
**To All**

We sincerely thank all the reviewers for their valuable time and insightful comments on our paper. We have carefully addressed all the suggestions and incorporated the changes in the main manuscript as well as the Appendix section. These modifications are highlighted in red within the revised manuscript for clarity.

---

### Meta-Review · Area_Chair_5rRQ · 2024-12-20

**Metareview:**

The paper proposes a new framework for video generation and dives into many of the important subparts: video autoencoding, diffusion training, and diffusion architectures. In addition, the paper also proposes a new downstream task (Sketch Guided Video Inpainting).

For video autoencoding, the paper proposes a new style of VAE (3D MoBile Inverted VQ-VAE) which trains with random masking of frames with patches and complete masking of a single frame. For diffusion training and architecture, the paper proposes to use discrete diffusion with Spectral Transformer blocks which use 2D FFT and Rotary Positional Embeddings. For the downstream task, the paper proposes Sketch-guided video inpainting where a user specifies with a sketch what the inpainted region should look like and create a dataset for this task. The paper demonstrates, for the VAE, strong reconstruction with decent compression; for the text to video model good quality (with interesting ablations on the design choices here); and nice results for the proposed sketch-based video inpainting task.

Reviewers all rated the paper above acceptance (two 8s, one 6) and particularly appreciated the work on the VAE and spectral transformer as well as the ablations on the design of the transformer and the new task.

Most of the weaknesses brought up by the reviewers (e.g. missing ablations / comparisons) were either addressed in the rebuttal process or seem to be "nice to haves" as opposed to "necessary" elements.

I agree with the reviewers and recommend acceptance. Note to the authors, integrating the rebuttal work (e.g. the image video joint training, masking ablation, etc.) with the paper would strengthen it and I strongly encourage the authors to do so.

**Additional Comments On Reviewer Discussion:**

The reviewer discussion primarily consisted of several clarifications and the identification of several additional ablations or experiments which the authors provided. For example:

- CyQT asked for an ablation on the masking strategy for the VAE which was provided by the authors.
- UAh7 asked for an an experiment on joint image video training which the authors provided; and an ablation on the effect of the codebook size which the authors provided.
- htH2 asked for an ablation on the loss function for the VAE which the authors provided.

---

### Decision · Program_Chairs · 2025-01-22

Accept (Spotlight)